# The RNA m5C modification in R-loops as an off switch of Alt-NHEJ

Haibo Yang[1,2], Emily M. Lachtara[1,3], Xiaojuan Ran[1,3,4], Jessica Hopkins[1,3], Parasvi S. Patel[1,3], Xueping Zhu[5], Yao Xiao[1,2], Laiyee Phoon[6], Boya Gao[1,2], Lee Zou[1,3,4], Michael S. Lawrence ●[1,3] & Li Lan ●[1,2,6] ✉

The roles of R-loops and RNA modifications in homologous recombination (HR) and other DNA double-stranded break (DSB) repair pathways remain poorly understood. Here, we find that DNA damage-induced RNA methyl-5-cytosine (m5C) modification in R-loops plays a crucial role to regulate PARP1-mediated poly ADP-ribosylation (PARylation) and the choice of DSB repair pathways at sites of R-loops. Through bisulfite sequencing, we discover that the methyltransferase TRDMT1 preferentially generates m5C after DNA damage in R-loops across the genome. In the absence of m5C, R-loops activate PARP1-mediated PARylation both in vitro and in cells. Concurrently, m5C promotes transcription-coupled HR (TC-HR) while suppressing PARP1-dependent alternative non-homologous end joining (Alt-NHEJ), favoring TC-HR over Alt-NHEJ in transcribed regions as the preferred repair pathway. Importantly, simultaneous disruption of both TC-HR and Alt-NHEJ with TRDMT1 and PARP or Polymerase θ inhibitors prevents alternative DSB repair and exhibits synergistic cytotoxic effects on cancer cells, suggesting an effective strategy to exploit genomic instability in cancer therapy.

Elevated replicative and oxidative stress is a common feature observed in many different types of cancer[1]. Recent studies, including our own, have shown that oxidative DNA damage and DNA double-stranded breaks (DSBs) in transcriptionally active regions of the genome[2–7] promote DNA:RNA hybridization and subsequent R-loop formation. Pathological R-loops are known to be susceptible to breakage[8,9], drive mutations[10,11], promote translocation[12], and may interfere with DNA replication[13,14], thus they pose a threat to genomic stability. On the other hand, R-loops play a beneficial role in transcription by protecting the DNA at the promoter region from methylation[15,16] and assisting in transcription termination[17–19]. Accumulating evidence shows that R-loops induced by DNA damage in a transcribed region of the genome act to facilitate the initiation of homologous recombination repair (HR)[5,20,21]. It has been shown that RNA modifications (e.g. methyl-6-

adenine (m6A) and methyl-5-cytosine (m5C)) occur at transcriptionally active DNA damage sites[22,23]. These modifications could promote DNA repair, but little is known whether or not DNA damage induced RNA modifications are widely distributed in the genome within specific sequence and structural contexts. While targeting R-loop modifiers and RNA modification enzymes is an attractive strategy in cancer therapy[24], further investigation is warranted to understand the molecular mechanisms by which RNA modifications regulate DNA repair.

DNA repair-targeted cancer therapy is increasingly used in the clinic, showing efficacy in tumors with elevated genomic instability. For example, PARP1/2 inhibitors (PARPi), which block PARP1/2 mediated poly-ADP-ribosylation (PARylation) and trap PARP1/2 on DNA, have been used in around 10–20% of patients with breast, ovarian and other cancers with BRCA1/2 mutations or homologous recombination

[1]Massachusetts General Hospital Cancer Center, Harvard Medical School, Boston, MA, USA. [2]Department of Radiation Oncology, Massachusetts General Hospital, Harvard Medical School, Boston, MA, USA. [3]Department of Pathology, Massachusetts General Hospital, Harvard Medical School, Boston, MA, USA. [4]Department of Pharmacology and Cancer Biology, Duke University School of Medicine, Durham, NC, USA. [5]Center for Immunology and Inflammatory Diseases, Division of Rheumatology, Allergy, and Immunology, Massachusetts General Hospital, Harvard Medical School, Boston, MA, USA. [6]Departments of Molecular Genetics and Microbiology, School of Medicine, Duke University, Durham, NC, USA. ✉e-mail: li.lan@duke.edu

(HR) deficiency (HRD)[25–27]. However, patients treated with PARPi inevitably develop drug resistance[28]. Our previous studies showed that TRDMT1-mediated RNA m5C modification promotes transcription coupled-homologous recombination (TC-HR) activity in cancer cells[5,22,29]. Several TC-HR proteins including TRDMT1 are commonly upregulated in breast and ovarian tumors, regardless of the BRCA status, and contribute to PARPi resistance in cancer cells[6,29], suggesting TRDMT1 as a promising therapeutic target for cancer treatment. PARP-mediated PARylation is well-known to serve as a docking platform for DNA repair factors and promote repair pathways including base excision repair (BER), alternative NHEJ (Alt-NHEJ) and others[30,31]. While both RNA modifications and PARylation play vital roles in DNA repair[30], the interplay between them at sites of DNA damage remains unknown.

In this study, we use bisulfite sequencing to show that TRDMT1-dependent m5C formation primarily occurs in R-loops, including DNA damage-induced R-loops, throughout the genome. We show that PARP1 is activated by R-loops, but the m5C-modified RNA:DNA hybrids prevent PARP activation both in vitro and in cells. Moreover, the m5C in damage-induced R-loops promotes TC-HR and simultaneously suppresses the PARP1-mediated alt-NHEJ, ensuring that TC-HR is the preferred DSB repair pathway in transcribed regions. The catalytic activity of TRDMT1 is essential for m5C-mediated PARP1 suppression. Loss of TRDMT1 reduces m5C and allows alt-NHEJ to occur, showing that alt-NHEJ can act as a backup pathway for TC-HR to repair DSBs in transcribed regions. Simultaneous disruption of TC-HR and alt-NHEJ with TRDMT1 inhibitor and PARP or Polymerase θ (POLθ) inhibitor kills cancer cells synergistically, providing a promising strategy to enhance the efficacy of PARP, POLθ and TRDMT1 targeted therapies.

## Results

### TRDMT1-mediated RNA m5C formation primarily occurs in R-loops in the genome

Recent studies suggest that DNA damage-induced RNA modifications, including the m5C modification, promote TC-HR and contribute to cancer cell survival[22]. However, the distribution of m5C in the genome remains elusive. Furthermore, whether m5C formation has any sequence or structural preferences has not been studied at a genome scale. To determine the distribution of TRDMT1-mediated m5C modification in the genome, we used the mRNA isolated from U2OS WT and TRDMT1 KO cells to perform bisulfite sequencing (Fig. 1a left panel)[32]. Because TRDMT1-mediated m5C formation is stimulated by DNA damage, cells were treated with $H_2O_2$ to enhance the modification (Supplementary Fig. 1a). The bisulfite treatment converts cytosine (C) residues to uracil (U), but leaves m5C residues intact. The remaining m5C in mRNA after bisulfite treatment was mapped by sequencing in WT and TRDMT1 KO cells. The sites of m5C detected in WT cells but not in TRDMT1 KO cells were designated as sites of the TRDMT1-dependent m5C modification events. We identified several hundred up to thousand TRDMT1-dependent and -independent m5C sites, which are distributed widely across the genome (Supplementary Fig. 1b). Specifically, we performed RNA m5C sequencing using bisulfite sequencing with and without $H_2O_2$ treatment in both WT cells and TRDMT1 KO cells. We observed a basal level of RNA m5C in cells without any damage exposure. Upon induction of DNA damage, we observed an increase in the levels of m5C in RNA. This finding suggests that DNA damage leads to an elevation in RNA m5C levels. In TRDMT1 KO cells, both before and after damage induction, we only detected an equivalent basal level of RNA m5C. This demonstrates that the induction of TRDMT1-dependent RNA m5C is specifically triggered by DNA damage (Fig. 1a middle panel), which is also observed in the mRNA m5C dot blot analysis[22] (Supplementary Fig. 2a). DNA-RNA immunoprecipitation (DRIP)-sequencing (DRIP-seq) has been widely used for genome-wide profiling of R-loops[33]. Since we previously showed that TRDMT1 induces m5C formation in damage-induced R-loops at a transcriptionally active locus[22], we analyzed the overlap between the

best m5C sites and DRIP-seq peaks (Fig. 1a). To focus on the R-loops of high confidence, we only included the RNaseH-sensitive DRIP-seq peaks in our analysis (Supplementary Fig. 2b, c). Strikingly, nearly half of the best 10% TRDMT1-dependent m5C sites were contained in the DRIP-seq peaks, which is significantly higher than best TRDMT1-independent m5C sites or unmethylated C sites contained in the DRIP-seq peaks (Supplementary Fig. 2d). The total TRDMT1-dependent m5C sites also revealed higher overlap percentage with the DRIP-peaks compared to that of the unmethylated C sites (Supplementary Fig. 2e). These results suggest that TRDMT1-mediated m5C formation preferentially occurs in R-loops genome-wide.

### TRDMT1 preferentially modifies Cs in Guanine (G)-rich RNA within R-loops

Next, we analyzed the sequence preference of the TRDMT1-dependent m5C sites. Motif analysis of the m5C sites revealed that Cs flanked by G-rich sequences are preferentially modified by TRDMT1 (Fig. 1b). We noticed that G enrichment diminishes as the distance from the m5C sites increases (Fig. 1b). Previous reports showed that GC rich sequences are R-loop prone[15,34]. The "GC skew" occurs when G and C are under- or over-abundant in a particular region of DNA or RNA strand, which affects R-loop formation in vitro and in cells[15,34]. It is known that G-rich RNA transcripts efficiently hybridize with the C-rich template (−) DNA strand, which exhibits high thermodynamic stability and stabilizes R-loops (Fig. 1c)[35]. Due to this thermodynamic stability, R-loops are preferentially formed in sequences where the sense (+) DNA strand is G-rich and the templated (−) stand is C-rich. In sequences around the most abundant m5C sites, we observed a clear G-bias in the sense (+) DNA strand (Fig. 1d). In contrast, the least abundant m5C sites do not show this sequence feature (Fig. 1d). A positive value of GC skew percentage represents a G-rich environment, while a negative value of GC skew percentage represents a C-rich environment. At the genome level, the analysis of DRIP-seq peaks indeed confirmed the G-bias sequence for the sense (+) strand and C-bias sequence for the template (−) strand (Fig. 1e). It should be noted that as the RNA secondary structure, which is easily formed in GC rich region, could affect the correct identification of m5C sites[36], we incorporated heat denature before the bisulfite conversion to disrupt the secondary structures. The distribution of both R-loops and TRDMT1-dependent m5C sites tend to exist in sequences with overall high GC content (X-axes in Fig. 1e, f). TRDMT1-dependent m5C sites presented a G-bias for the sense (+) strand and a C-bias for the template (−) strand as reflected by GC skew analysis (Y-axis in Fig. 1f). Together, our results suggest that TRDMT1-dependent m5C formation preferentially occurs in G-rich RNA within R-loops.

Notably, the DRIP-seq data were generated from undamaged cells, whereas the TRDMT1-dependent m5C sites were mapped in $H_2O_2$-treated cells. Given that TRDMT1-mediated m5C formation is induced at DNA damage sites[5,22], some of the m5C sites detected by bisulfite sequencing may be associated with damage-induced R-loops. We performed DRIP-qPCR on several loci in the cells treated with or without $H_2O_2$. Three types of loci were analyzed based on m5C mapping and DRIP-seq results: a control site without m5C and a DRIP-seq peak (Ctrl), a site with m5C overlapping with a DRIP-seq peak (m5C-R), and three sites with m5C but no DRIP-seq peaks (m5C1; m5C2; and m5C3). The levels of DNA:RNA hybrids at these loci were compared to that of the Ctrl locus (Fig. 1g). Consistent with the DRIP-seq data, the DNA:RNA hybrid levels at the m5C-R locus were higher than that of the Ctrl locus (Fig. 1g left). Importantly, in $H_2O_2$-treated cells, the DNA:RNA hybrid levels at the m5C1, m5C2, and m5C3 loci were also higher than that of the Ctrl locus, suggesting that these loci contain damage-induced R-loops (Fig. 1g left). Indeed, at the m5C2 and m5C3 loci, the levels of DNA:RNA hybrids increased significantly after DNA damage (Fig. 1g right). The RNaseH

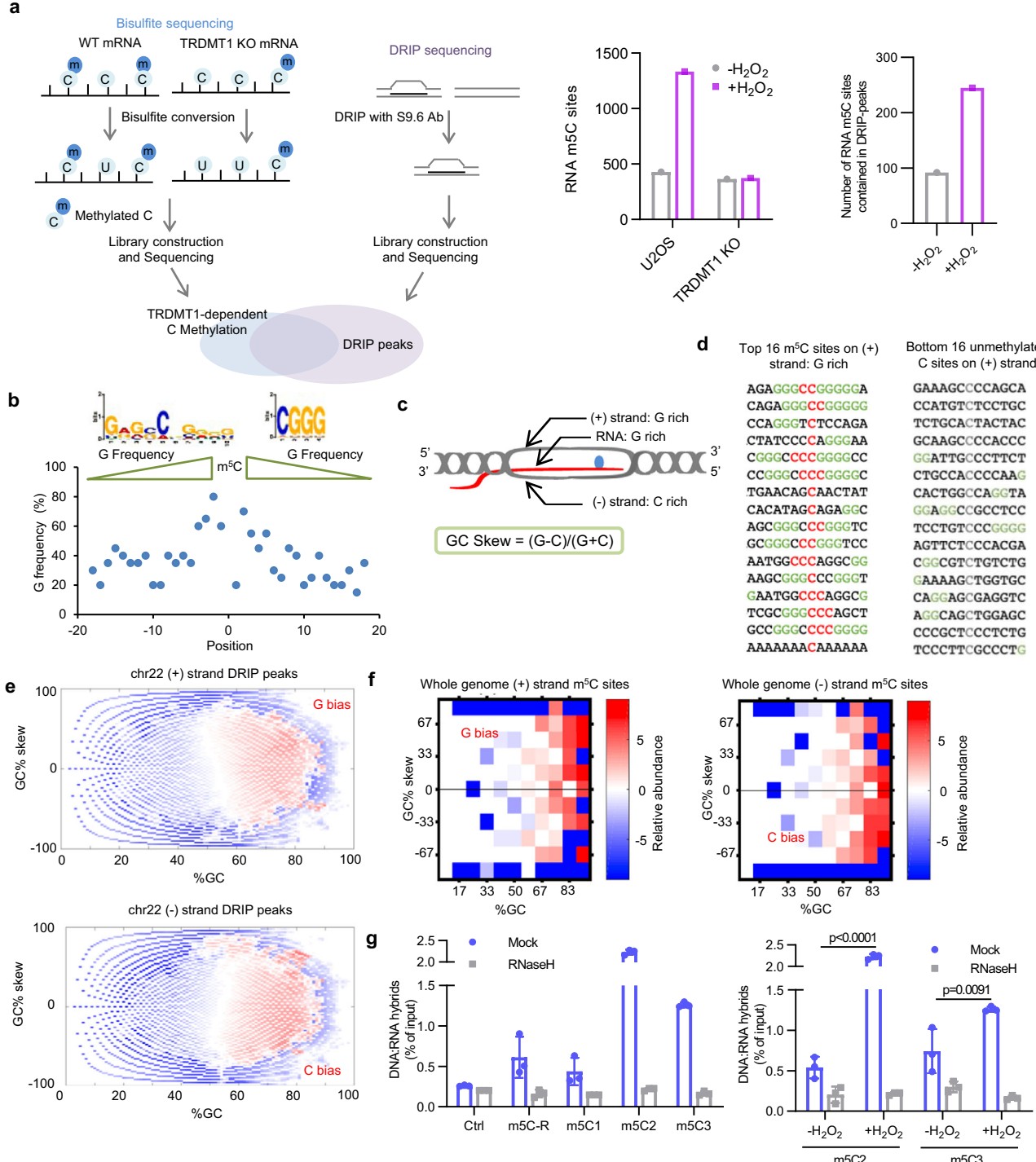

**Fig. 1 | TRDMT1 mediated m5C RNA modifications enriched in G bias context were contained in R-loop with high GC content. a** Left panel: Schematic diagram of the bisulfite sequencing to identify the TRDMT1-dependent or -independent m5C RNA modifications sites and DRIP sequencing to identify DNA:RNA hybrid regions. Middle panel: The number of RNA m5C sites identified form bisulfite sequencing. Right panel: The number of RNA m5C sites contained in the DRIP-seq peaks. **b** The sequence context of top 20 TRDMT1-dependent m5C sites. The G frequency was plotted. **c** Schematic diagram of the R-loop structure with features of different strands. **d** Sequence features of the top and bottom methylated sites on + strands. **e** Observed (R-loops) vs Expected (the remainder of the genome) ratios were calculated for each coordinate of GC skew (x) vs GC fraction (y). Coordinates where the observed was enriched compared to the expected are red, and

coordinates where the observed was depleted compared to the expected are blue. Plots were faceted by transcribing strands. **f** The distribution of whole genome m5C sites was plotted based on both GC content and GC skew. Plots were faceted by transcribing strands. **g** The levels of DNA:RNA hybrids in different locus were analyzed by DRIP-qPCR. The U2OS cells were treated with or without 1 mM $H_2O_2$ for 1 h before harvest. The RNaseH treatment was included before DRIP reaction. The analyzed loci are with different feature of m5C and DRIP peak. Data are presented as mean ± SD ($n = 3$ independent experiments). Three types of loci were used: a control site without m5C or a DRIP peak (Ctrl), a site with m5C overlapping a DRIP peak (m5C-R), three sites with m5C but not DRIP peaks (m5C1; m5C2; and m5C3). Statistical analysis was done with one-way ANOVA. Source data are provided as a Source Data file.

treatment before the DRIP reaction decreases the DNA:RNA hybrid to the basal level, indicating DRIP-qPCR at m5C-R, m5C1, m5C2 and m5C3 reflects the real DNA:RNA hybrid level. Moreover, at the genome-wide level, the RNA m5C sites identified in U2OS WT cells from bisulfite sequencing revealed an increased overlap with the DRIP-seq peaks after $H_2O_2$ damage induction (Fig. 1a right panel). Together, these results suggest that TRDMT1 promotes m5C formation in both constitutive and damage-induced R-loops.

## RNA m5C modification prevents PARP1 activation by DNA:RNA hybrids in vitro

To understand the role of RNA m5C modification within G-rich R-loops, we synthesized biotin-labeled, G-rich, single-stand RNA (ssRNA) oligos with or without m5C based on the sequence of an abundant m5C site in cells. Furthermore, we annealed these ssRNA oligos with complementary single-strand DNA (ssDNA) to generate DNA:RNA hybrids. The DNA:RNA hybrids with or without m5Cs were incubated with lysates of HEK293 cells treated with $H_2O_2$, and pulled down by Streptavidin. The proteins pulled down by the empty beads control, unmodified or m5C-modified hybrids were analyzed by mass spectrometry (Fig. 2a). We identified ~100–200 proteins with unmodified or m5C-modified hybrids. A KEGG enrichment analysis of the binding proteins revealed that proteins involved in DNA repair and RNA biogenesis pathways are enriched (Supplementary Fig. 3a). Among these proteins, some, such as the RNA binding protein YBX3, bound to unmodified and m5C-modified hybrids similarly. Some transcription factors, such as SMAD9 and ENY2, preferentially bound to m5C-modified hybrids (Fig. 2b). To our surprise, PARP1 was one of the top proteins that preferentially bound to unmodified hybrids. DNA repair factors XRCC1 and Ligase 3 (LIG3) were also preferentially bound to unmodified hybrids compared to control and modified hybrids (Fig. 2b, Supplementary Table 1). These results indicate that PARP1 is capable of binding DNA:RNA hybrids, but its binding may be inhibited by m5C. PARP1 is known to have a high affinity to double-stranded DNA (dsDNA)[37,38]. To directly test the binding of PARP1 to DNA:RNA hybrids, we performed electrophoretic mobility shift assay (EMSA) with 50-bp hybrids, and used 50-bp dsDNA as a positive control. We observed robust PARP1 binding to DNA:RNA hybrids with a binding affinity >50% of that for dsDNA (Fig. 2c). Therefore, PARP1 can bind DNA:RNA hybrids efficiently in vitro. Surprisingly, the DNA:RNA hybrid with m5C displayed a similar affinity to PARP1 as the unmodified DNA;RNA hybrid (Fig. 2d), suggesting that m5C does not directly affect PARP1 binding to DNA:RNA hybrids in vitro. However, it should be noted that PARP1 activation promotes the accumulation of PARP1 on DNA through a feed-forward loop[39], raising the possibility that hybrids with or without m5C may activate PARP1 differently in cells or cell extracts and indirectly affect their association with PARP1. This possibility prompted us to investigate whether the m5C in hybrid affects PARP1 activation.

Upon binding to DNA breaks, PARP1 is activated to synthesize poly(ADP-ribose) (PAR) at sites of DNA damage to promote DNA repair[30,31]. Because PARP1 can bind DNA:RNA hybrids, we performed in vitro PARP1 activation assay to check whether DNA:RNA hybrids could activate PARP1. In the presence of PARP1, NAD+ and dsDNA (Fig. 2e Lane 5), abundant PAR signals were observed as expected. PARP1 alone was not sufficient for PAR formation (Fig. 2e Lane 1), confirming that this assay is NAD+ and DNA-dependent. PAR signals were detected when dsDNA was replaced by DNA:RNA hybrids (Fig. 2e Lane 3), and these PAR signals were also dependent on NAD+ (Fig. 2e Lane 2). Thus, although m5C-modified hybrids still bind PARP1, they did not trigger PARylation as efficiently as unmodified hybrids or

dsDNA (Fig. 2e Lane 4). These results suggest that m5C prevents the efficient activation of PARP1 by DNA:RNA hybrids.

## RNA m5C modification suppresses the activation of PARP1 at R-loops in cells

Having shown that m5C RNA modifications may prevent PARP1 activities in vitro, we sought to investigate the effect of transcription, R-loops, and the RNA m5C modification on the recruitment of PARP1 to DNA damage sites in cells. We used DNA damage at RNA transcribed sites (DART) assay to compare PARP1 recruitment in transcriptionally on and off states[4,40]. In the DART assay, KillerRed (KR), a light excitable chromophore, releases free radicals upon light activation in a dose-dependent manner. A cassette containing an array of tandem tetracycline responsive elements and an adjacent reporter gene was stably integrated into the genome of U2OS cells. KR is fused with either tetR (tetR-KR) or tetR and the transcription activator VP16 (TA-KR) and expressed in cells. The fusion proteins tetR-KR and TA-KR are recruited to the TRE locus. TetR-KR binds to the TRE locus but does not activate transcription, whereas TA-KR binds and activates the reporter gene transcription locally (Fig. 3a). After KR activation with visible light exposure, TetR-KR and TA-KR release free radicals and cause the same levels of DSBs at the TRE locus (Supplementary Fig. 4a)[40]. However, robust recruitment of PARP1 was preferentially detected in cells expressing tetR-KR but not TA-KR (Fig. 3a), indicating that the recruitment of PARP1 to damage sites was inhibited by transcription. PARP1 recruitment to the TRE site bound by tetR-KR is consistent with the activation of PARP1 by DNA breaks[37,38]. As we showed previously, co-transcriptional R-loops are induced by TA-KR but not tetR-KR (Supplementary Fig. 4b), and the R-loops induced by TA-KR are sensitive to RNaseH treatment (Supplementary Fig. 4c)[5]. Furthermore, m5C was induced at the TRE site by TA-KR but not tetR-KR (Fig. 3a)[22]. Thus, PARP1 is recruited to DNA damage sites when transcription is off and in the absence of damage-induced R-loops and RNA m5C modification.

Our in vitro results suggest that PARP1 binds DNA:RNA hybrids and synthesizes PAR. Moreover, m5C rather than DNA:RNA hybrids prevent PARP1 activation (Fig. 2). To distinguish the effects of R-loops and m5C on the recruitment of PARP1 in cells, we next tested the recruitment of PARP1 to the TRE site bound by TA-KR in TRDMT1 knockout (KO) cells. In TRDMT1 KO cells, R-loops were normally induced by TA-KR, but the damage-induced m5C formation was reduced more than 2-fold (Fig. 3b)[22]. Importantly, the recruitment of PARP1 to the TRE site bound by TA-KR in TRDMT1 KO cells was significantly increased compared to that in U2OS WT cells (Fig. 3c), suggesting that m5C is responsible for the exclusion of PARP1 from DNA breaks. Given that m5C does not directly inhibit the binding of PARP1 to DNA:RNA hybrids but prevents the activation of PARP1 by hybrids in vitro (Fig. 2d, e), our result suggests that m5C may prevent PARP1 recruitment by inhibiting PARP1 activation and disrupting the PAR-driven feed-forward loop that promotes PARP1 accumulation[39]. To test this possibility, we examined the PARylation level at m5C-modified R-loops in U2OS WT and unmodified R-loops in TRDMT1 KO cells. PARylation levels at unmodified R-loops in TRDMT1 KO are higher than those at m5C-modified R-loops in U2OS WT cells (Fig. 3d). Together, these results suggest that RNA m5C modification but not the damage-induced R-loops interfere with PARP1 activation at DNA breaks in transcribed regions. The increased PARP1 recruitment and activation in TRDMT1 KO cells might benefit PARP1-stimulated DNA repair pathways at DNA damage sites.

## The catalytic activity of TRDMT1 is required for repressing PARP1 activation in cells

To further investigate whether the TRDMT1-mediated m5C formation prevents PARP1 activation by DNA damage, we expressed TRDMT1 WT or catalytically inactive TRDMT1 mutants in TRDMT1

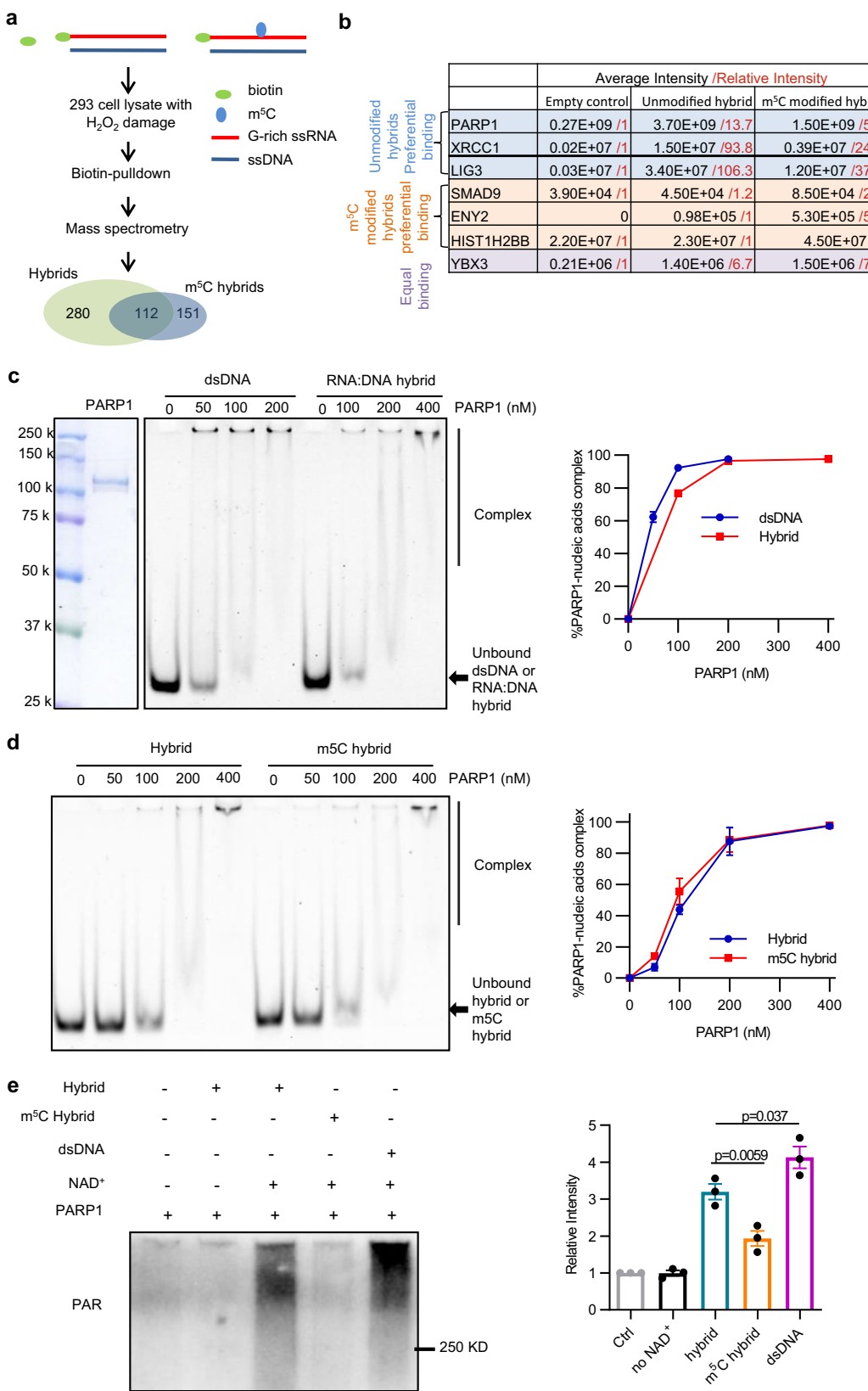

KO cells. TRDMT1 C79A is defective in both RNA binding and catalyzing m5C formation, and R162A is only defective in catalyzing m5C formation[22]. TRDMT1 WT, C79A, and R162A were expressed at similar levels in TRDMT1 KO cells (Fig. 4a). TRDMT1 WT repressed the recruitment of PARP1 to the TRE site damaged by TA-KR, but the catalytically inactive mutants C79A and R162A did not (Fig. 4b).

Similarly, TRDMT1 WT repressed PARylation at the damaged site, but C79A and R162A did not (Fig. 4c). We also examined global PARylation levels in lysates from cells treated with or without low dose ionizing radiation (IR). IR leads to increased PARylation in both U2OS WT and TRDMT1 KO cells (Fig. 4d). In TRDMT1 KO cells, only the expression of TRDMT1 WT, but not C79A or R162A, repressed IR-

**Fig. 2 | m5C RNA modifications within the RNA:DNA hybrid suppress the PARP1 activity. a** Schematic diagram of mass spectrometry to identify binding partners of RNA:DNA hybrid with or without m5C RNA modifications modification. **b** Example of binding partners showing preferentially binding to unmodified RNA:DNA hybrid or RNA:DNA hybrid with m5C RNA modifications. **c** The binding of PARP1 protein to unmodified RNA:DNA hybrids was measured in electrophoretic mobility shift assays. The dsDNA with same sequence was used as a positive control. Data are presented as mean ± SD ($n = 3$ independent experiments). The Kd values for the binding of PARP1 with dsDNA and RNA:DNA hybrid are determined to be 2.25 nM

and 4 nM, respectively. **d** The binding of PARP1 protein to RNA:DNA hybrids with or without m5C RNA modifications was measured in electrophoretic mobility shift assays. Data are presented as mean ± SD ($n = 3$ independent experiments). **e** In vitro PARP1 activation assay with purified PARP1, NAD + , and RNA:DNA hybrid with or without m5C RNA modification. Same concentration of dsDNA was used as positive control. Data are presented as mean ± SD ($n = 3$ independent experiments). Statistical analysis was done with one-way ANOVA. Source data are provided as a Source Data file.

induced global PARylation (Fig. 4d). Thus, the catalytic activity of TRDMT1 to form m5C is required for repressing PARP1 recruitment and activation by DNA damage.

## TRDMT1 deficiency promotes PARP1-mediated Alt-NHEJ

Loss of TRDMT1 leads to decreased TC-HR without affecting KU70/Ku80-dependent NHEJ[22]. Apart from HR and NHEJ, Alt-NHEJ also contributes to the repair of DSBs[41]. Notably, PARP1 is a key factor in the Alt-NHEJ pathway[42]. Our result that TRDMT1-dependent m5C formation at R-loops inhibits PARP1 activation raised the possibility that PARP1-mediated Alt-NHEJ is also regulated by TRDMT1. To test this possibility, we used the EJ2-GFP reporter to analyze the Alt-NHEJ activity in U2OS WT and TRDMT1 KO cells[43]. The Alt-NHEJ activity in TRDMT1 KO cells was higher than that in U2OS WT cells (Fig. 5a, Supplementary Fig. 5a), supporting the idea that TRDMT1 inhibits Alt-NHEJ. In addition to PARP1, XRCC1, LIG3 and POLθ are required for the repair synthesis and ligation during Alt-NHEJ[41] (Fig. 5b). We examined recruitment of these factors to the TRE site damaged by TA-KR. POLθ was increasingly recruited to the damaged site in TRDMT1 KO cells compared to U2OS WT cells (Fig. 5c). Likewise, the recruitment of XRCC1 and LIG3 to the damaged site in TRDMT1 KO cells was also higher than that in U2OS WT cells (Fig. 5d, e). Our results suggest that TRDMT1-induced m5C inhibits the recruitment of PARP1-dependent Alt-NHEJ factors and the activity of this pathway. To address whether the upregulation of Alt-NHEJ in TRDMT1 deficient cells is truly dependent on PARP1, we tested the effects of the PARP inhibitor Olaparib on the EJ2-GFP reporter in TRDMT1 KO cells. While TRDMT1 KO increased Alt-NHEJ repair (Fig. 5a, f), this increase was completely reversed by PARPi (Fig. 5f). Thus, TRDMT1 deficiency increases Alt-NHEJ in a PARP-dependent manner. The effects of TRDMT1 mutants in the EJ2-GFP reporter assay were also tested. While TRDMT1 WT restored the suppression of alt-NHEJ repair, C79A and R162A did not (Fig. 5g), reinforcing the catalytic activity of TRDMT1 to form m5C is required for suppression of alt-NHEJ. Moreover, to assess the effect of TRDMT1 on single strand annealing (SSA), another repair pathway of DSBs[44], we examined the efficiency of the SSA repair in TRDMT1 KO cells using the SA-GFP reporter assay[45] (Fig. 5h). The SSA repair efficiency is unchanged compared to WT cells. We further investigated factors known to affect the DNA damage repair pathway choice. The protein levels of RNA m6A writers[23], METTL3 and METTL14, are unaffected in TRDMT1 KO compared to U2OS WT (Supplementary Fig. 5b), indicating TRDMT1 loss does not affect level of m6A writers. Moreover, we didn't detect m6A at transcribed (TA-KR) or nontranscribed regions (tetR-KR) of the genome, both before and after light activation of KR induced DNA damage[22]. Additionally, we evaluated the protein levels of cell cycle proteins, e.g. Cyclin B1, Cyclin E, P53[46], and TGFβ[47]. The western blot results indicate that the protein levels of these factors remain unchanged in TRDMT1 KO cells compared to U2OS WT cells (Supplementary Fig. 5c). Consequently, the aforementioned results collectively suggest that the shift towards alt-NHEJ repair in TRDMT1 KO cells is not mediated by changes in m6A modification or cell cycle regulation.

## TRDMT1 inhibitor synergizes with PARPi or Polθi in killing HR-proficient cancer cells

Previous studies suggested that RAD52 is essential for the survival of BRCA1/2 deficient cancer cells[48,49], suggesting that a RAD52-dependent repair pathway is indispensable in BRCA1/2-deficient cells. We previously showed that RAD52 recognizes R-loops and plays an important role in the TRDMT1 and m5C-regulated TC-HR pathway[22]. PARP1 recruitment to the TA-KR damaged site was not affected by RAD52 or RAD51 KD (Supplementary Fig. 6), suggesting that the downstream TC-HR events mediated by RAD52 and RAD51 are not involved in PARP1 inhibition. This observation further supports the notion that TRDMT1 inhibits PARP1 activation through m5C rather than downstream TC-HR. Notably, the TRDMT1-mediated TC-HR pathway can operate at sites of ROS-induced DNA damage independently of BRCA1/2[5,22], suggesting that TRDMT1 inhibition may impair DSB repair even in BRCA1/2-deficient cells. To test this possibility, we tested the effects of YW-1842, a TRDMT1 inhibitor (TRDMT1i), in control and BRCA1 knockdown cells[29]. BRCA1 depletion in the HR-proficient breast cancer cell line HS578T significantly increased TRDMT1i sensitivity (Fig. 6a), showing that TRDMT1 indeed has a BRCA1-independent function that is critical for the survival of BRCA1-deficient cells.

Our data suggest that TRDMT1 inhibition should block TC-HR but allow Alt-NHEJ to occur. Similar to TRDMT1 knockdown, TRDMT1i indeed increased Alt-NHEJ (Supplementary Fig. 7a). Therefore, Alt-NHEJ may at least partially compensate for the loss of TC-HR in DSB repair upon TRDMT1 inhibition, contributing to the damage resistance of cancer cells. According to this idea, one would predict that inhibition of both TRDMT1 and Alt-NHEJ should kill cancer cells more effectively. Furthermore, even in BRCA1/2-proficient cancer cells, blocking two DSB repair pathways may exploit genomic instability more efficiently and lead to cell death. To test this possibility, we treated the BRCA1/2-proficient breast cancer cell lines HS578T and MDA-MB-231 with TRDMT1i and either PARPi (Olaparib) or POLθi (Novobiocin) (Fig. 6b, c)[50]. The combinations of TRDMT1i with PARPi or POLθi killed the HR-proficient cancer cells much more efficiently than single drugs alone (Fig. 6d, e). Notably, when used in combinations, the IC50s of TRDMT1i in HS578T and MDA-MB-231 cells were well below 1 µM, which was significantly lower than the IC50s of TRDMT1i when used alone (Fig. 6f, g). The combination of TRDMT1i and PARPi induced DNA damage robustly in cancer cells as shown by γH2AX foci (Supplementary Fig. 7b), while the cell cycle distribution was not affected (Supplementary Fig. 7c). Thus, by blocking TC-HR and Alt-NHEJ simultaneously, the combination of TRDMT1i with PARPi or POLθi induces a "synthetic lethality" in HR-proficient cancer cells (Fig. 6h).

## Discussion

### How does RNA m5C RNA modification prevent PARP1 activation?

PARP1 can be activated by different DNA structures, including single- and double-strand breaks[38]. Here we show the PARP1 binds and is activated by DNA:RNA hybrids. The roles of damage-induced R-loops in DNA repair are increasingly appreciated[8], and the activation of PARP1 by DNA:RNA hybrids could add another layer of regulation to DNA repair. Actually, the association of PARP1 with R-loops has just

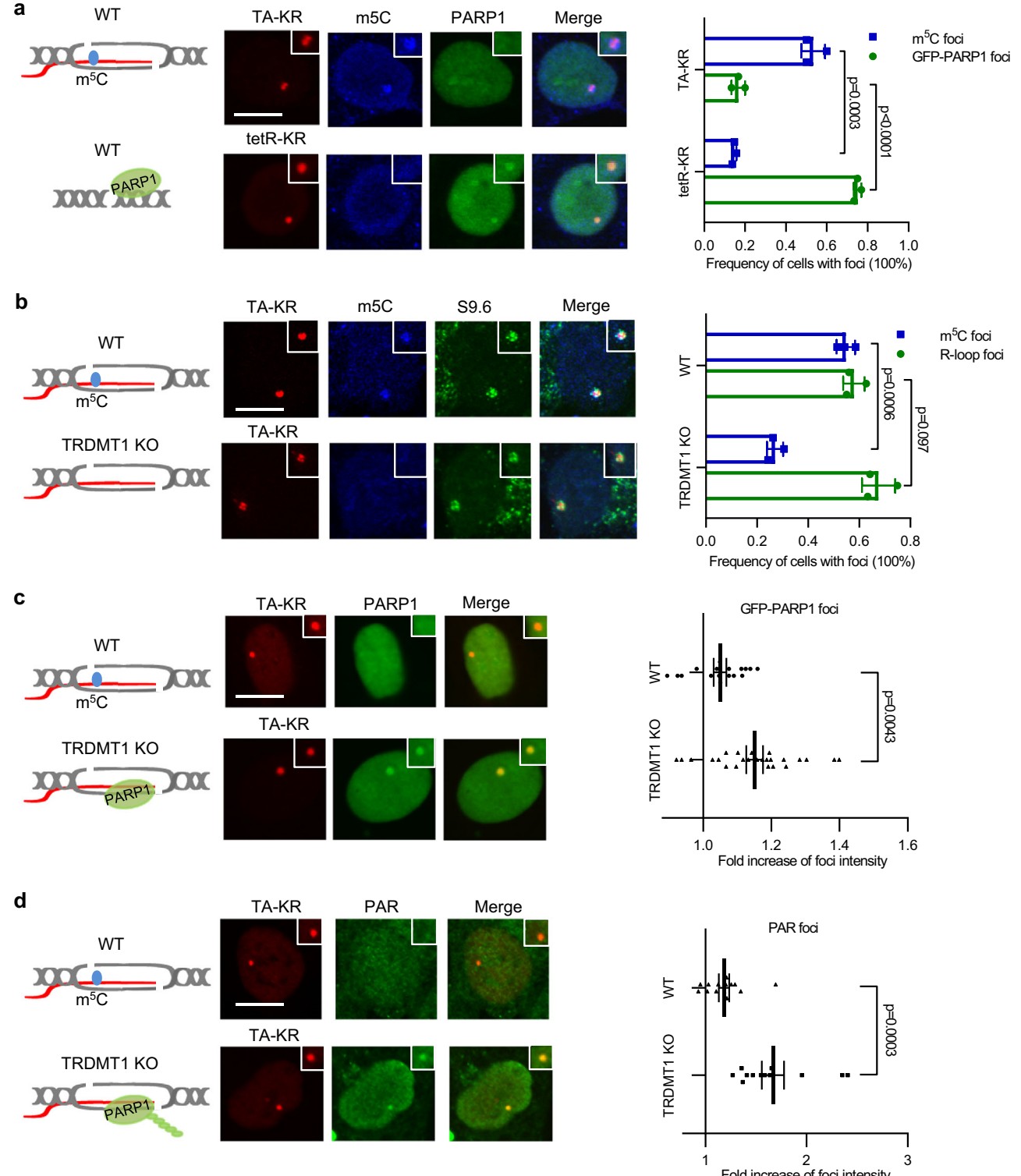

**Fig. 3 | m5C RNA modifications suppress the damage response of PARP1 at transcribed damaging sites. a** U2OS-TRE cells transfected with TA-KR/tet-KR and GFP-PARP1 plasmids were light irradiated and allowed to recover for 0.5 h (h) before fixation. Cells were stained with m5C antibody (scale bar: 10 μm). RNA m5C foci frequency and GFP-PARP1 foci frequency were quantified ($n = 3$ independent experiments, 50 cells per replicate, Mean ± SD). **b** WT and TRDMT1 KO U2OS-TRE cells transfected with TA-KR plasmid were light irradiated and allowed to recover for 0.5 h before fixation. Cells were stained with m5C and S9.6 antibody (scale bar: 10 μm). RNA m5C foci frequency and R-loop foci frequency were quantified ($n = 3$, 50 cells per replicate, Mean ± SD). **c** WT and TRDMT1 KO U2OS-TRE cells transfected with TA-KR and GFP-PARP1 plasmids were light irradiated and allowed to recover for 0.5 h before fixation (scale bar: 10 μm). Fold increase of GFP-PARP1 foci intensity was quantified. Mean intensity of PARP1 at TA-KR /mean intensity of background is shown ($n = 17$ cells for WT, $n = 25$ cells for TRDMT1 KO, Mean ± SEM). **d** WT and TRDMT1 KO U2OS-TRE cells transfected with TA-KR plasmid were light irradiated and allowed to recover for 0.5 h before fixation. Cells were stained with PAR antibody (scale bar: 10 μm). Fold increase of PAR foci intensity was quantified. Mean intensity of PAR at TA-KR /mean intensity of background is shown ($n = 12$ cells, Mean ± SEM). Statistical analysis was done with the unpaired two-tailed student t-test. Source data are provided as a Source Data file.

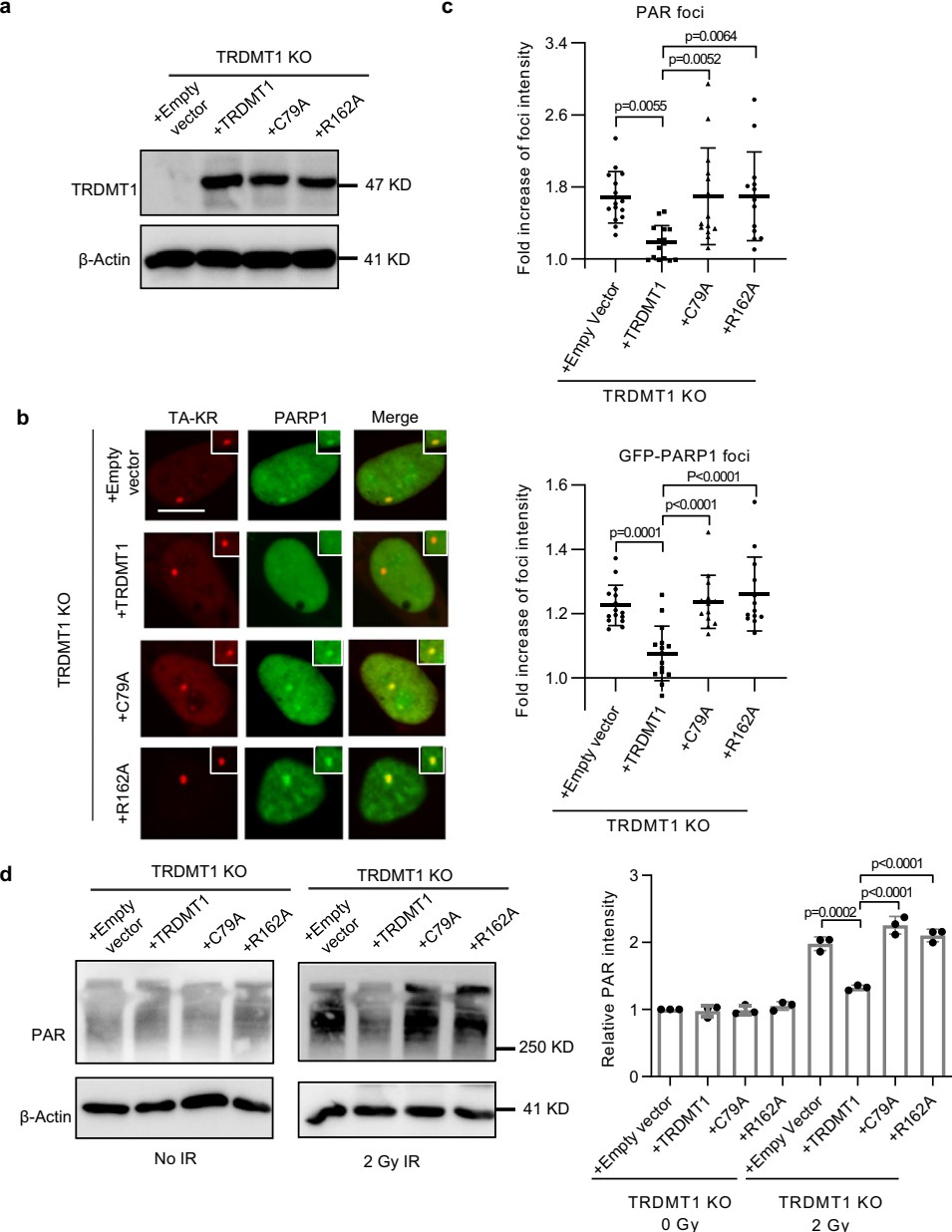

**Fig. 4 | The catalytic activity of TRDMT1 is vital for the suppression of PARP1 activation. a** Stable expression of Myc-tagged TRDMT1 in TRDMT1 KO U2OS-TRE cell lines shown in Western blots. Expression of β-actin is shown as control. The experiments were repeated independently three times with similar results. **b** U2OS-TRE TRDMT1 stably expressing cells transfected with TA-KR and GFP-PARP1 plasmids were light irradiated and allowed to recover for 0.5 h before fixation (scale bar: 10 μm). Fold increase of GFP-PARP1 foci intensity was quantified. Mean intensity of PARP1 at TA-KR /mean intensity of background is shown (*n* = 13 cells, Mean ± SEM). **c** U2OS-TRE TRDMT1 stably expressing cells transfected with TA-KR plasmid were light irradiated and allowed to recover for 0.5 h before fixation. Cells were stained with PAR antibody. Fold increase of PAR foci intensity was quantified. Mean intensity of PAR at TA-KR /mean intensity of background is shown (*n* = 13 cells, Mean ± SEM). **d** U2OS-TRE TRDMT1 stably expressing cells were irradiated with 2 Gy IR and were allowed for recover for 0.5 h. Cells with or without IR were collected and analyzed by Western blot for PAR level. The levels of PAR was quantified (*n* = 3 independent experiments, Mean ± SD). Statistical analysis was done with one-way ANOVA. Source data are provided as a Source Data file.

been reported and the association could trigger the PARylation of PARP1[51]. Furthermore, we demonstrate that the m5Cs in RNA:DNA hybrids inhibit the recruitment and activation of PARP1 in cells, but do not alter the hybrid binding of PARP1 in vitro. The binding of PARP1 to damaged DNA is enhanced by a positive feedback loop driven by PARP1-mediated PARylation[39]. It is possible that the initial binding of PARP1 to the m5C-modified hybrids is not affected, but m5Cs prevents PARylation and the subsequent amplification of PARP1 recruitment to DNA damage sites in cells. The mechanism underlying the suppression of PARP1 activation by m5Cs in DNA:RNA hybrids warrants further

investigations. It is known that the BRCT domain of PARP1 could prevent the enzymatic activation of PARP1 if BRCT is engaged with intact DNA[52]. One possibility is that m5Cs facilitate the engagement of hybrids with the BRCT, which would then inhibit the activation of PARP1. Alternatively, m5Cs may prevent the unfolding of the helical domain (HD) of PARP1, which is a key event in PARP1 activation that opens the active site and allows productive binding of NAD+[53,54]. Finally, certain m5C readers, such as RAD52 and FMRP[22,55], may compete with PARP1 for binding with R-loop, thus reducing the activation of PARP1 by R-loops.

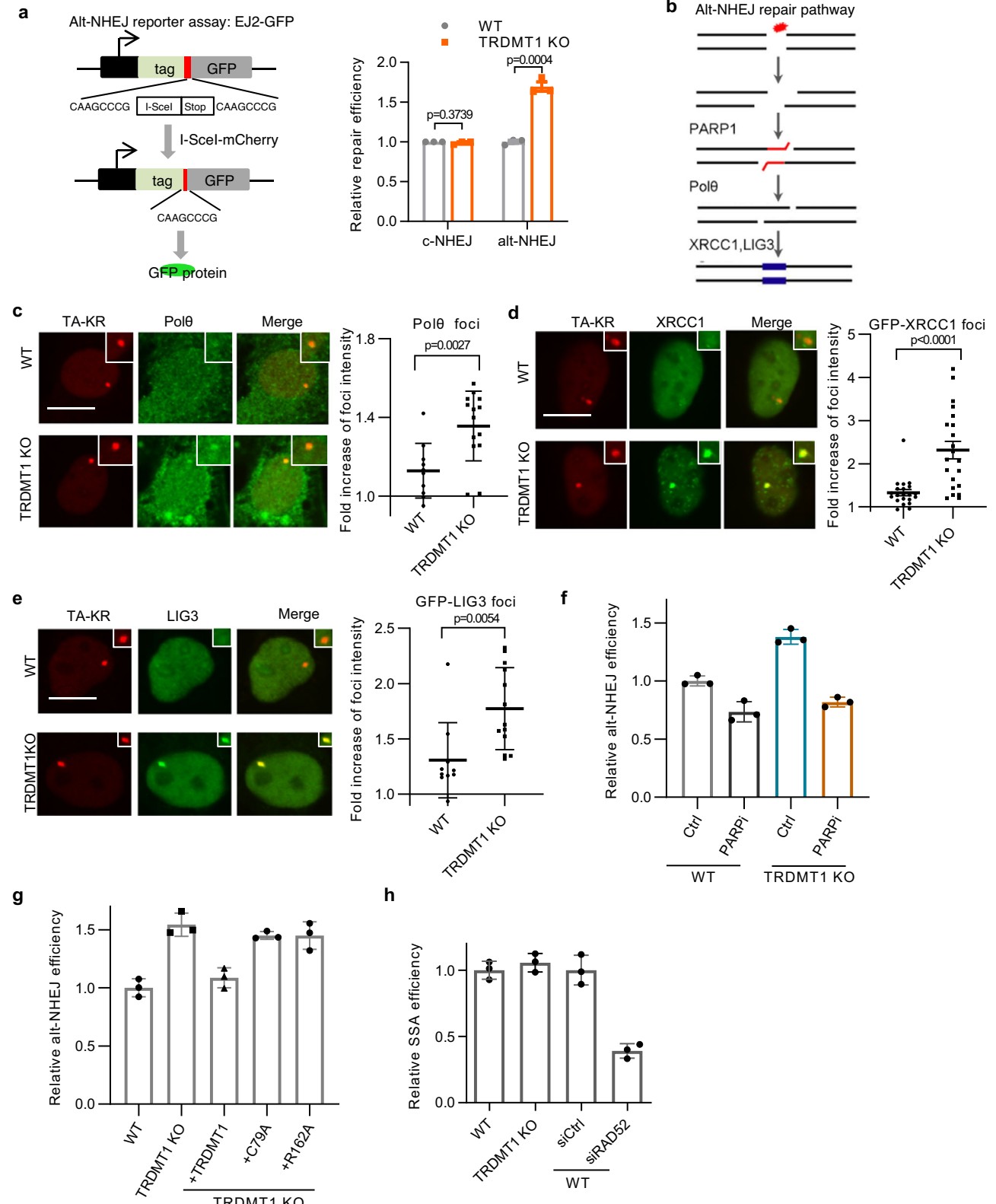

## How is the DNA repair pathway choice regulated by m5C?

Our previous study showed that TRDMT1 mediated RNA m5C modification promotes TC-HR[22]. In this study, we show that TRDMT1 mediated m5C formation suppresses PARP1 activation in the DNA:RNA hybrids at transcribed damage sites, and represses the recruitment of downstream Alt-NHEJ factors, which leads to the suppression of DSB repair by Alt-NHEJ. Specifically, we found that m5C formation, facilitated by TRDMT1, suppresses the activation of PARP1 in DNA:RNA hybrids at sites where damage has occurred during transcription. It is worth noting that NHEJ and SSA pathways are not affected by RNA m5C. This suppression of PARP1 activation leads to a decrease in the recruitment of downstream factors involved in Alt-NHEJ pathway. As a result, the repair of DNA double-strand breaks (DSBs) through Alt-NHEJ is suppressed. Importantly, the absence of m5C modification,

**Fig. 5 | Deficiency of TRDMT1 promotes the alt-NHEJ repair. a** WT and TRDMT1 KO U2OS-TRE cells were transfected with EJ2-GFP/EJ5-GFP and I-SceI-Cherry plasmids. The fraction of GFP-positive cells in the Cherry-positive population was analyzed by flow cytometry (n = 3 independent experiments, Mean ± SD). **b** Schematic diagram of the alt-NHEJ repair pathway. **c–e** WT and TRDMT1 KO U2OS-TRE cells transfected with TA-KR plasmid were light irradiated and allowed to recover for 0.5 h before fixation. Representative images are showed (scale bar: 10 μm). Fold increase of foci intensity was quantified and mean intensity of foci at TA-KR /mean intensity of background is shown. Cells were stained with Polθ antibody (n = 10 cells, Mean ± SEM) **c**, or cells were co-transfected with GFP-XRCC1 plasmid (n = 21 cells, Mean ± SEM) **d** or GFP-LIG3 plasmid (n = 10 cells, Mean ± SEM)

**e. f, g** U2OS-TRE cells were transfected with EJ2-GFP and I-SceI-Cherry plasmids. The fraction of GFP-positive cells in the Cherry-positive population was analyzed by flow cytometry (n = 3 independent experiments, Mean ± SD). Before and after plasmids transfection, the cells were also treated with or without 3 μM PARPi Olaparib **f. h** WT and TRDMT1 KO U2OS-TRE cells pretreated with or without RAD52 siRNA were transfected with SA-GFP and I-SceI-Cherry plasmids. The fraction of GFP-positive cells in the Cherry-positive population was analyzed by flow cytometry (n = 3 independent experiments, Mean ± SD). Statistical analysis was done with the unpaired two-tailed student t-test, ns: not significant. Source data are provided as a Source Data file.

caused by the lack of TRDMT1 activity, leads to the activation of Alt-NHEJ at the transcribed damage sites, suggesting Alt-NHEJ serves as a primary backup repair at sites of damage-induced R-loops. While the influence of transcription on DNA repair has been recognized for some time, the role of RNA and its modifications in DNA repair is still being unraveled. For example, the incorporation of ribonucleotides at the DNA break ends promotes the repair by NHEJ, both during the repair of chromosome breaks made by Cas9 and during V(D)J recombination[56]. Furthermore, several studies have suggested that transcription promotes HR at the DSBs induced by DNA endonucleases or oxidative damage[2–5]. Additionally, the RNAs induced by DSBs, including small RNAs and long non-coding RNAs, have also been shown to promote DSB repair[20,21,57,58]. Our findings contribute to the understanding of a new regulatory mechanism involving R-loops (DNA:RNA hybrids) and RNA m5C modification in the choice between different DNA repair pathways. Specifically, this mechanism includes the accurate TC-HR pathway and a backup Alt-NHEJ pathway in transcribed regions, thereby safeguarding genomic integrity and contributing to cell survival.

Finally, the combination of TRDMT1i with PARPi or Polθi, which blocks both TC- HR and Alt-NHEJ, kills HR-proficient breast cancer cells efficiently. Recently, combination therapies involving PARP inhibitors have been extensively explored and have shown promising results in various clinical settings. Combination therapies involving PARP inhibitors can take different forms, such as combining PARP inhibitors with chemotherapy, radiation therapy, immunotherapy, or other targeted agents. These combinations are designed to exploit synergistic interactions between different treatment modalities, potentially leading to improved outcomes and overcoming resistance mechanisms[59]. In this study, we reason that cancer cells with genomic instability or repair defects are particularly sensitive to the loss of both TC-HR and Alt-NHEJ pathways, providing the basis for the combination therapy using TRDMT1i and PARPi or Polθi. In addition to breast and ovarian cancers, TC-HR is upregulated in lung, prostate, sarcoma, blood, pancreatic, and cervical cancers[29]. Thus, targeting the TC-HR pathway may broadly improve the treatment of a wide range of cancers. TRDMT1i are expected to be effective in BRCA1/2-deficient tumors, and also overcome PARPi/platinum resistance by blocking BRCA1/2-independent DSB repair. Further investigations are warranted to test how TRDMT1i affects DNA repair, cancer cell survival, and tumor growth. These studies could lead to therapies targeting TC-HR and broaden the exploitation of genomic instability in both BRCA1/2-deficient and proficient tumors.

## Methods
### Cell culture and transfection
U2OS TRE[40], TRDMT1 KO U2OS TRE[22], Flp-in 293 (Thermo, Cat#R75007), HS578T and MDA-MB-231 cells were cultured in Dulbecco's modified Eagle medium (DMEM, Cat#12−604 F, Lonza; Basel, Switzerland) with 10% (vol/vol) FBS at 37 °C with 5% CO$_2$. The U2OS-TRE cells used for the DART system have been described in previous articles[40]. The HS578T and MDA-MB-231 cells are gifts from Dr. Leif W. Ellisen. For plasmid and siRNA transfection, Lipofectamine 2000 and

Lipofectamine RNAiMax (Invitrogen; Carlsbad, CA, USA) were used following the manufacturer's standard protocol, respectively. The siRNA for TRDMT1 was purchased from Invitrogen (siRNA ID: s4219, Cat#: 4392420). Other siRNAs include siBRCA1 (L-003461-00, Dharmacon), and siBRCA2 (GS675, Qiagen).

### Bisulfite sequencing
WT and TRDMT1 KO U2OS TRE cells treated with or without 1 mM H$_2$O$_2$ for 1 h were collected and total RNA was extracted using the Trizol. The mRNA was purified with a Dynabeads™ mRNA DIRECT™ Purification Kit (Cat#: 61011, ThermoFisher Scientific, Waltham, MA, USA) from the total RNA. After the removal of DNA with a DNA removal kit, the mRNA was heated for 5 mins and then snapped cool down on ice before applied to bisulfite conversion. The conversion was conducted with the EZ RNA methylation kit (Zymo Research). The spike-in RNA (UUAAUUGGGUGUGACUAAUCGAAGUUGAUA-CAUCGACGUUAUGGUGAUGAU) with mass ratio of 1:40000 were used. After conversion, the RNA was sent to Novogene (Sacramento, CA, USA) for sequencing.

### DNA−RNA immunoprecipitation (DRIP)-seq
DNA−RNA immunoprecipitation DNA−RNA immunoprecipitation (DRIP) assays were performed as described previously (Sanz and Chédin 2019). Briefly, genomic DNA was extracted from wildtype U2OS cells by SDS/proteinase K treatment overnight at 37 °C, followed by phenol-chloroform extraction and ethanol precipitation at room temperature. Genomic DNA was subsequently digested for 24 h at 37 °C using a cocktail of restriction enzymes containing BsrGI (New England Biolabs Cat # R3575L), EcoRI (New England Biolabs Cat # R3101L), HindIII (New England Biolabs Cat # R3104L), SspI (New England Biolabs Cat # R3132L), and XbaI (New England Biolabs Cat # R0145L) in 1x NEB Buffer 2 (Cat # B7002) supplemented with 1X BSA (New England Biolabs Cat # B9000) and 2 mM spermidine (Sigma Cat #05292-1ML-F). Digested samples were re-purified to remove restriction enzymes, followed by treatment with or without thermostable RNASEH enzyme for 6–8 h at 37 °C in NEB RNase H buffer (New England Biolabs Cat # M0523). Next, DNA−RNA hybrids were immunoprecipitated 16 h overnight at 4 °C using monoclonal S9.6 antibody (Antibodies Incorporated) diluted in IP buffer with continuous rotation, followed by 4 h incubation at 4 °C with 50uL Pierce™ protein A/G magnetic beads (Thermo Fisher Cat# 88803). Antibody-coupled magnetic beads were washed gently three times with IP buffer (10 min per wash) prior to release of immunoprecipitated nucleic acids in elution buffer for 45 min at 55 °C. Immunoprecipitated hybrids were then subjected to phenol−chloroform extraction followed by ethanol precipitation, resuspension in 0.1xTE and RNase A digestion (New England Biolabs Cat # T3018L) for 1 h at 37 °C. Samples were then sonicated (Qsonica Q800R) and fragmented DNAs were double-size selected to retain fragments between 200−600 bp using SPRIselect beads (Beckman Cat # B23318). Unique dual-index barcodes were then ligated onto the eluted fragments according to the manufacturer's protocol (NEBNext® Ultra™ II DNA Library Prep Kit for Illumina® Cat # E7645; Dual Index Multiplex Oligos Cat # E7600), and the resulting

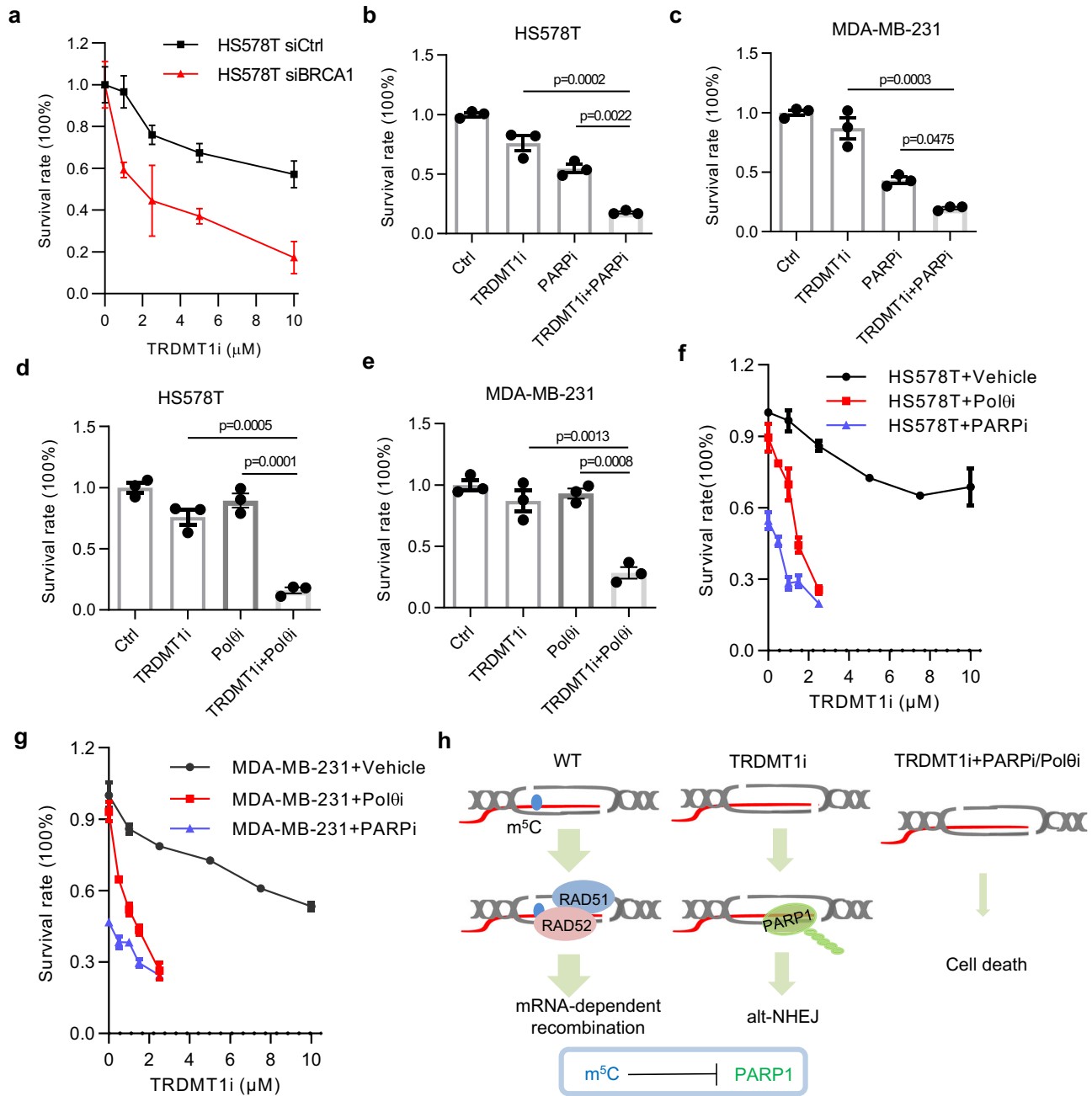

**Fig. 6 | TRDMT1 inhibitor kills breast cancer cells in combination with PARPi or Polθi. a** HS578T cells pretreated with siBRCA1, or control siRNA were treated with different concentration of TRDMT1 inhibitor, YW-1842 and were cultured for 7 days. Survival rate was measured via colony formation assay (*n* = 3 independent experiments, Mean ± SEM). **b, c** HS578T cells **b** or MDA-MB-231 cells **c** were treated with 2.5 μM YW-1842 or 1 μM PARPi Olaparib were cultured for 7 days. Survival rate was measured via colony formation assay (*n* = 3 independent experiments, Mean ± SEM). **d, e** HS578T cells **d** or MDA-MB-231 cells **e** were treated with 2.5 μM YW-1842 or 50 μM Polθi Novbiocin were cultured for 7 days. Survival rate was measured via colony formation assay (*n* = 3 independent experiments, Mean ± SEM). **f, g** HS578T cells **f** or MDA-MB-231 cells **g** were treated with different concentration of YW-1842 combined with 1 μM PARPi Olaparib or 50 μM Polθi Novbiocin and were cultured for 7 days. Survival rate was measured via colony formation assay (*n* = 3 independent experiments, Mean ± SEM). **h** A scheme shows the different repair pathways for the DSB at transcribed region in WT and TRDMT1 KO cells. Statistical analysis was done with one-way ANOVA. Source data are provided as a Source Data file.

DRIP-seq libraries were subjected to next-generation sequencing (NovaSeq 6000).

## Computational analysis
The raw data was processed using samtools and bamutils to remove duplicated reads and overlapping regions. The meRanTK toolkit was used to call methylation sites and identify m5Cs from the data (meRanCall). TRDMT1-dependent sites were selected on the basis of having a two-fold greater percentage of methylation than the TRDMT1-independent sites, and vice versa. The best m5C sites were determined on the basis of high coverage (>25 reads) and high methylation (>10 %).

Consensus peaks were called using Macs2 (determine where R-loops form) in DRIP-seq and qDRIP-seq data derived from U2OS WT cells. Peaks that were also present in U2OS cells treated with RNaseH were removed from analysis. A m5C site was considered "in an R-loop" if it resided in one of these DRIP-seq peaks.

GC skew and GC fraction were calculated for the entire genome on a 50 bp window. Observed (R-loops) vs Expected (the remainder of the genome) ratios were calculated for each coordinate of GC skew (x) vs GC fraction (y). Coordinates, where the observed was enriched compared to the expected, are red, and coordinates where the observed was depleted compared to the expected are blue. Plots were faceted by transcribing strands.

### DRIP-qPCR assay

The DRIP assay was performed according to previous literature (15). U2OS TRE cells in a 10 cm dish were treated with 1 mM H2O2 for 1 h before harvest. The cells were trypsinized and resuspended in 1.6 ml of TE with 0.5% of SDS and 5 μl of Proteinase K (Roche Life Sciences), then the cells were incubated overnight at 37 °C. The genomic DNA was extracted with phenol/chloroform in MaXtract High Density phase lock gel tubes (Qiagen), and precipitated with ethanol/sodium acetate. After washing 5 times with 70% ethanol, the genomic DNA was resuspended in TE. DNA was digested with HindIII, BsrGI, XbaI, EcoRI, and SspI at 37 °C overnight and was purified by phenol/chloroform and ethanol method. Digested genomic DNA was treated with or without RNaseH1 (Cat#: EN0201, ThermoFisher Scientific) for 5 h at 37 °C. Then, 4.4 μg DNA was bound with 10 μg of S9.6 antibody in 1× DRIP binding buffer (10 mM NaPO$_4$ pH7.0, 140 mM NaCl, 0.05% Triton X-100) overnight at 4 °C. Washed protein G agarose beads were added for an additional 2 h. After washing three times in 1× DRIP binding buffer, the bound immunocomplexes were eluted in Elution Buffer (50 mM Tris pH 8.0, 10 mM EDTA, 0.5% SDS, Proteinase K) at 55 °C with rotation. The eluted samples were then subjected to nucleic acid purification using the phenol/chloroform and ethanol method. The immunoprecipitated nucleic acid and input DNA was analyzed by qPCR using Forward (F) and Reverse (R) primers: CtrlF: AAAGCATCAGCACCAGAACGC, CtrlR: CGAGGAAGGGACCCAATAACC; m5C-RF: GTGCCATCACTCAACCATAACA, m5C-RR: TTCCTGCCTGCTAGAAATCATC; m5C1F: AGAACTCGTTCCCGAATGTGC, m5C1R: TGGGACCATCAAATATCATAGCC; m5C2F: TCTACGATGGCTCCTGGGTGG, m5C2R: GCAGCCAACAAGTCAACGGA; m5C3F: CCAGTTCTCCAGGTGCGTTAC, m5C3R: GTCCTCCTCAGCAGTTAGTTAT. The analyzed loci with different feature of m5C and DRIP peak were selected based on the analysis result of bisulfite sequencing and DRIP sequencing.

### In vitro RNA:DNA hybrid pulldown and MS analysis

3′-biotin labeled RNA oligonucleotides with or without m5C (ssRNA, 5′-GGGCCGCCGUAGCXUCCACGGGGCACCGGC-3′, X = C or m5C) and complementary ssDNA (5′- GCCGGTGCCCCGTGGAGGCTACGGCGGCCC-3′) were synthesized from IDT. The ssRNA with or without m5C modification was annealed with ssDNA in a 1:1 ratio to generate the RNA:DNA hybrid.

Flp-in 293 cells treated with 1 mM H$_2$O$_2$ for 1 h were collected and suspended in lysis buffer (50 mM Tris-HCl pH 7.5, 150 mM NaCl, 0.4 mM EDTA, 1%NP-40, 0.4 U/μL RNasin inhibitor). After 30 min incubation on ice, cell lysate was centrifuged at 16000 x g for 15 mins. The up-layer supernatants were precleared by pre-washed streptavidin-conjugated magnetic beads for 1 h at 4 °C. The streptavidin-conjugated magnetic beads were also precleared by 0.2 mg/ml tRNA and 0.2 mg/ml BSA for 1 h at 4 °C. The precleared beads were incubated with empty control, or RNA:DNA hybrid, or m5C modified RNA:DNA hybrid in RNA binding buffer (50 mM Tris–HCl pH7.5, 150 mM NaCl, 0.5% NP-40, 10 mM MgCl$_2$) for 0.5 h at 4 °C. After washing with RNA binding buffer for 3 times, the beads-nucleic acids complex were incubated with the precleared cell lysate for 0.5 h at room temperature and then at 4 °C for 2 h. After washing with lysate buffer 3 times, the beads-nucleic acids-protein mixture was heated in SDS loading buffer at 95 °C for 5 min, then subjected to electrophoresis in 10% SDS-PAGE. The gel slices were sent to Harvard Medical School Taplin Biological Mass Spectrometry Facility for MS analysis.

Excised gel bands were cut into approximately 1 mm3 pieces. Gel pieces were then subjected to a modified in-gel trypsin digestion procedure. Gel pieces were washed and dehydrated with acetonitrile for 10 min. followed by removal of acetonitrile. Pieces were then completely dried in a speed-vac. Rehydration of the gel pieces was with 50 mM ammonium bicarbonate solution containing 12.5 ng/μl modified sequencing-grade trypsin (Promega, Madison, WI) at 4 °C. After 45 min., the excess trypsin solution was removed and replaced with 50 mM ammonium bicarbonate solution to just cover the gel pieces. Samples were then placed in a 37 °C room overnight. Peptides were later extracted by removing the ammonium bicarbonate solution, followed by one wash with a solution containing 50% acetonitrile and 1% formic acid. The extracts were then dried in a speed-vac (~1 h). The samples were then stored at 4 °C until analysis. On the day of analysis the samples were reconstituted in 5–10 μl of HPLC solvent A (2.5% acetonitrile, 0.1% formic acid). A nano-scale reverse-phase HPLC capillary column was created by packing 2.6 μm C18 spherical silica beads into a fused silica capillary (100 μm inner diameter x ~30 cm length) with a flame-drawn tip. After equilibrating the column each sample was loaded via a Famos auto sampler (LC Packings, San Francisco CA) onto the column. A gradient was formed and peptides were eluted with increasing concentrations of solvent B (97.5% acetonitrile, 0.1% formic acid). As peptides eluted they were subjected to electrospray ionization and then entered into a Velos Orbitrap Elite ion-trap mass spectrometer (Thermo Fisher Scientific, Waltham, MA). Peptides were detected, isolated, and fragmented to produce a tandem mass spectrum of specific fragment ions for each peptide. Peptide sequences (and hence protein identity) were determined by matching protein databases with the acquired fragmentation pattern by the software program, Sequest[60] (Thermo Fisher Scientific, Waltham, MA). All databases include a reversed version of all the sequences and the data was filtered to between a one and two percent peptide false discovery rate. The type of mass spectrometer is Velos Orbitrap Elite ion-trap mass spectrometer (Thermo Fisher Scientific, Waltham, MA). MS acquisition settings include MS1 m/z range: 360–1250, MS1 resolution: 120000, MS2 resolution: low resolution in the ion trap, Top20 data dependent acquisition, dynamic exclusion on, reject single charge states. Gas phase fragmentation settings include Collision Induced Disassociation (CID) fragmentation, isolation width of 2 m/z. The databases have been downloaded from Uniprot.

### Preparation of hybrid substrate used for EMSA

The 5′ ends of oligos were labeled using a 5′ oligonucleotide end-labeling kit (Vector Laboratories; Burlingame, CA, USA) and maleimide-IR800 probe (LI-COR Bioscience; Lincoln, NE, USA). RNA-DNA or m5C DNA:RNA hybrid substrates were prepared similarly to the literature[61]. Briefly, 5′ end-labeled oligo 1 was mixed either with oligo 2 or oligo 3 or oligo 4 (Supplementary Table 2) in buffer H (Tris-HCl [pH 7.5] 90 mM, MgCl$_2$ 10 mM, NaCl 50 mM), heat denatured and annealed by slow cooling. Annealed substrates were separated by 10% native PAGE-TAE. The corresponding gel bands were excised and eluted.

### Electrophoretic mobility shift assay

5′ maleimide-IR800-labeled substrates (20 nM) were incubated with PARP1 (Abcam, Catalog # ab123834) in binding buffer (20 mM Tris-HCl [pH 7.5], 50 mM KCl, 1 mM DTT) for 20 min at 37 °C. Reactions were loaded on a 6% PAGE-TBE gel and resolved at 4 °C. Gels were imaged using the BIO-RAD (Hercules, CA, USA) ChemiDoc™ MP imaging system.

## Plasmids

The TA-KR, tetR-KR on pBroad3 plasmids, GFP-PARP1, GFP-XRCC1 and GFP-LigIII plasmids used for the DART system have been described[40]. The plasmids used for alt-NHEJ reporter assay include I-SceI-mCherry and EJ2-GFP. The Myc-taged TRDMT1 WT, C79A, R162A plasmids were used for the re-expression of TRDMT1 in TRDMT1 pre-depleting cells[22].

## Microscopy and activation of KR

The Olympus FV1000 confocal microscopy system (Cat#: F10PRDMYR-1, Olympus; Waltham, MA, USA) and FV1000 software were used for acquisition of images. Cells were cultured in 35-mm glass-bottom dishes (P35GC-1.5-14-C, MatTek; Ashland, MA, USA) before observation. Activation of KR in bulky cells was completed by exposing them to a 15-W Sylvania (Wilmington, MA, USA) cool white fluorescent bulb for 25 min in a UVP stage. The intensity was measured by ImageJ 1.50i software. *P* values were calculated by the Student's t test.

## Immunoassays and m⁵C staining

Cells for immunofluorescence observation were fixed in 4% PFA (19943 1 LT, Affymetrix/ThermoFisher Scientific) for 15 min at room temperature and further treated with 0.2% Triton X-100 for 8 min. They were then blocked by 5% BSA (A-7030, Sigma-Aldrich) for 1 h at room temperature. Primary antibodies were diluted in 5% BSA and incubated with cells overnight at 4 °C. The samples were then washed three times with 0.05% PBST, and the cells were incubated with secondary antibodies for 1 h at room temperature followed by three washes with 0.05% PBST. Incubation with (1:1,000 dilution) DAPI for 10 min at room temperature was optional. Antibodies used in this study are summarized in Supplementary Table 3.

For m⁵C and S9.6 staining using the heat method[62], cells were fixed and permeabilized in a 35-mm glass-bottom dish, incubated in buffer (10 mM Tris-HCl, 2 mM EDTA, pH 9), and steamed on a 95 °C heating block for 20 min to expose the antigen. The dish was cooled, washed three times by PBS and blocked using 5% BSA in 0.1% PBST for 1 h at room temperature. The primary and secondary antibodies were diluted in the same buffer (5% BSA in 0.1% PBST) and followed the standard IF protocol.

## Western blots

U2OS or Flp-in 293 cells were collected and suspended in lysis buffer (10 mM Hepes, pH 7.6, 50 mM Tris-HCl, pH 7.4, 150 mM NaCl, 1% NP-40, 5 mM EDTA, protease inhibitor from Roche). After 30 mins incubation on ice, cell lysate was centrifuged at 16000xg for 15 mins. The up-layer supernatants were boiled in SDS loading buffer at 95 °C for 5–10 min, and applied to electrophoresis with 10–12% SDS-polyacrylamide gels, followed by transferring to PVDF membranes. For block and antibody dilution, 5% non-fat milk in PBST was used. After primary antibody incubation at 4 °C overnight and secondary antibody incubation at room temperature for 1 h, the membranes were washed in 0.1% PBST three times, respectively. Chemiluminescent HRP substrate was purchased from Millipore (Cat#: WBKLS0500; Burlington, MA, USA). Images were taken in the BIO-RAD (Hercules, CA, USA) ChemiDoc™ MP imaging system.

## In vitro PARP1 activation assay

Recombinant PARP1 protein (1 nM) (Thermo Fisher, Catalog # A42572) were incubated in 20 μL PAR reaction buffer (100 mM Tris-HCl pH8, 10 mM MgCl₂, 10% glycerol, 10% ethanol and 10 mM DTT) with different types of nucleic acids, including dsDNA, DNA:RNA hybrids with or without m5C RNA modifications, which were generated by annealing different oligos (Supplementary Table 2). The PARylation reactions were started by adding 25 mM NAD⁺ to the reaction mixtures. After incubation for 10 min at room temperature, the reaction was immediately terminated by heating in SDS loading buffer at 95 °C for 5 min, followed by Western blot analysis.

## EJ2-GFP, EJ5-GFP, SA-GFP reporter assay and flow cytometry

U2OS cells were transfected with the linearized EJ2-GFP, EJ5-GFP, or SA-GFP. The I-SceI-mCherry plasmid was co-transfected as an internal control for transfection. Two days after transfection, the cells were collected for flow cytometry analysis. The normal cell population was gated in P1 by SSC-A and FSC-A. The repair efficiency was then calculated as the ratio of GFP-positive cell number to mCherry-positive cell number. The gating strategy was demonstrated in Supplementary Fig. 8.

## Cell survival assay

Approximately 400 cells were seeded in each 6-cm dish and cultured as described above. They were treated with different DNA damaging reagents after seeding. After 7–10 days, colonies were fixed and stained with 0.3% crystal violet in methanol, and the number of colonies was counted manually.

## Reporting summary

Further information on research design is available in the Nature Portfolio Reporting Summary linked to this article.

## Data availability

The bisulfite sequencing raw data has been deposited into NCBI's BioProject and is available through the accession numbers PRJNA833238 and PRJNA984695. The DRIP-seq raw data has been deposited into NCBI's BioProject with the accession number PRJNA833771. The DNA:RNA hybrid pulldown fraction mass spectrometry raw data and search files have been deposited into the MassIVE data repository and are accessible through the PDX identifier PXD045391. Source data are provided in this paper.

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

## Acknowledgements

This work was supported in part by grants from the National Institutes of Health to L.L. (GM118833 and CA282939).

## Author contributions

H.Y. conducted the major experiments. E.M.L. and X.R. performed the sequencing data analysis. J.H., P.S.P., X. Z., Y.X., L.P. and B.G. participated in the experiments and data analysis. M.S.L. and L.Z. participated in experimental design and data analysis. L.L. provided overall experimental guidance. All authors acknowledge their specific contributions and have reviewed the manuscript.

## Competing interests

The authors declare no competing interests.
