## [Peer Review File · Nature Communications]

The RNA m5C Modification in R-loops as an Off Switch of Alt-NHEJREVIEWER COMMENTS

Reviewer #1 (Remarks to the Author):

Yang et al. propose a follow up work starting from the premises of their 2020 paper, demonstrating that TRDMT1-mediated m5C RNA modification at transcribed loci not only promotes HR repair of DNA damage, but it also prevents PARP-mediated Alt-NHEJ engagement.

First, they investigate the localization of TRDMT1-dependent m5C modification relatively to R-loops formation and sequence context. Subsequently, they identify proteins that bind differentially to unmodified and m5C modified hybrids, and they discover PARP among them. They use their published cell model to demonstrate that PARP enrichment to the site of damage is prevented by TRDMT1-dependent RNA methylation, and this is reflected by a boost in alt-NHEJ engagement and alt-NHEJ factors recruitment to the site of DNA damage in the absence of TRDMT1. Finally, they observe an interesting synthetic lethality using TRDMT1 inhibitors and alt-NHEJ factors inhibitors in cancer cells. This is an overall interesting and solid paper, and I only have some concerns about the first set of results and the statistics used throughout the manuscript.

The authors propose that TRDMT1-mediated RNA m5C formation primarily occurs in R-loops in the genome. To support this claim, they combine data of bisulfite-seq and DRIP-seq in H₂O₂-treated and unchallenged cells respectively. Next, they check the formation of damage-induced R-loops 2 out of 3 loci investigated as positive for m5C but negative for DRIP signal. I find these data a bit weak to reach their conclusions, as the presence of R-loops and the deposition of m5C was never investigated in the same exact condition. DRIP-seq in H₂O₂-treated cells could be beneficial to support authors' point, ideally showing a more specific co-presence of m5C and R-loops after DNA damage induction.

Figure 1G: DRIP-qPCR experiment should include RNaseH negative controls to ensure that the differences in signal are representative of the modulation in hybrids formation

Figure S2C: it is unclear, in text authors report a comparison between the best TRDMT1-dependent m5C sites and the overall m5C, while, in the figure the bar is labeled as unspecified "Top 32 methylated C sites", and the caption reports "The fraction of top 32 methylated sites or bottom 32 unmethylated sites contained in the DRIP peaks".

Moreover, I am not certain that the comparison between the best TRDMT1-dependent m5C sites (defined in terms of coverage and methylation levels) and overall m5C is meaningful: the greater percentage of TRDMT1-dependent m5C sites colocalizing with DRIP peaks could be due to the fact that these "best sites" are more methylated (as they were selected based on that), rather than because they are TRDMT1-dependent.

Figure 2: in panel C and D authors could indicate the complexes of PARP+ substrate and the unbound substrate with arrows. Panel E lacks statistics

Figure 4: all the comparisons in the figure should be tested by ANOVA, not t-test, which is used when the comparison is only between two samples.

Figure 5: Authors could add a representative image also for Pol θ , as they did for XRCC1 and Lig3. Again, in figure 5F, the wrong statistical test was performed.

Figure 6: use of t-test instead of ANOVA throughout the figure

Reviewer #2 (Remarks to the Author):

Bisulfite sequencing has been widely used to identify m5C modification in DNA and RNA. However, this technique suffers some drawbacks mainly due to the fact that sodium bisulfite-mediated deamination of C to U occurs successfully only on ssRNA. Therefore, potentially false positive coming from RNA secondary and higher order structures (e.g. GC-rich regions) might affect the correct identification of m5C sites. Furthermore, due to technical artifacts and lack of proper control, the m5C mapped sites can widely range from 100 to 12000 in mammalian genome.

In this study, the identification of TRDMT1-dependent m5C sites has been determined in wt cells and TRDMT1 KO cells. However, the data from bisulfite sequencing should be repeated using catalytically inactive TRMDT1 mutants, since compensatory roles of other m5C-methyltransferases cannot be excluded at this stage. Furthermore, to exclude artifacts coming from GC-rich sequences, other m5C sequencing techniques such as TAWO-seq should be considered.

DRIP-seq employs the S9.6 antibody which binds RNA:DNA hybrids allowing the identification of the genomic fragments containing RNA:DNA hybrids. However, this technique has some crucial limitations including low resolution, no-strand specificity and the intrinsic bias of the S9.6 antibody toward GC-rich sequences (Bou-Nader Nat comm. 2022). Therefore, the analysis of the overlapping signal between the m5C and RNaseH-dependent RNA:DNA might be bias to the identification of m5C sites in GC-rich environment.

RNA modifications such as m5C and m6A have can both promote HR via the modulation of DNA:RNA hybrid stability. It should be crucial to characterize the expression profile of m6A-writers in TRDMT1 ko cells, since the lack of TRDMT1 activity might change the expression level of m6A-writers and the deposition of m6A at damage sites and the choice of HR over NHEJ repair pathway. No data about the regulation of cell cycle and cell cycle progression in wt and TRDMT1 KO have been provided (there are no controls provided for TRDMT1 inhibitor).

Furthermore, evaluation of down/up-regulated genes which can influence the choice of DNA damage repair pathway, cyclins expression, p53 and TGF-beta signaling is pivotal to claim the direct role of TRDMT1 and m5C role on the DNA repair pathway choice in vivo. Finally, since hydrogen peroxide can induce both ssDNA and dsDNA damage, would be informative show the gH2AX levels after H2O2 treatment.

Figure 2 shows the binding affinity of PARP1 in presence of dsDNA and DNA:RNA hybrids. It would beneficial to add the dissociation constant values of PARP1-dsDNA and PARP1-DNA:RNA hybrid. Furthermore, it is not clear whether other RNA-related damage modifications such as m6A and 8-OHG can influence PARP1 recruitment and PAR activity.

Figure 5 illustrates how the lack of TRDMT1 protein affects the choice of Alt-NHEJ over c-NHEJ. To confirm that TRDMT1-driven m5c deposition is responsible for the exclusion of PARP1 from DNA break in EJ2/EJ5-GFP reporter assays, TRDMT1 ko cells should be rescued with catalytically inactive C79A and R162A TRDMT1 mutants. Furthermore, DR-GFP reporter assay and SA-GFP reporter should be added to exclude the involvement of TRMDT1 in other DNA repair pathways. Also no controls for reporter assays were provided (KD of known related proteins).

To exclude indirect effects of the TRDMT1 absence on the switch off of Alt-NHEJ pathway further controls must be provided.

Reviewer #3 (Remarks to the Author):

This work by Yang and colleagues is a follow-up to a 2020 study that examines the role of m5C modification of RNA and the subsequent effect on DNA repair pathway choice. The group has identified TRDMT1 as the enzyme responsible for m5C modification. Here, they demonstrate a correlation with regions that form transcriptionally coupled R-loops and sites of m5C modification of RNA following hydrogen peroxide treatment, meaning that these are sites where DNA damage-induced R-loops are forming and TRDMT1 is involved with m5C modification of RNA in an RNA:DNA hybrid. In the previous work, it was show that this modification led to the recruitment of Rad52 to drive HR repair. Here, they

show that when the ability to generate m5C modifications is blocked via a TRDMT1 deletion, PARP1 is recruited to RNA:DNA hybrids and activation and PARylation commences, meaning that the 5mC modification driven by TRDMT1 is important in DSB repair pathway choice. It is suggested that pharmacological inhibition of TRDMT1 activity may drive cells to a PARP1-dependent repair pathway (alt-EJ) and subsequent inhibition of PARP will force the cells into a death pathway. While there are somewhat interesting results here implicating a novel factor with little-known prior function in a major repair pathway choice mechanism, it is difficult to evaluate the significance of the findings without better explanation of some of the results and additional experiments.

Specific comments:

1. For the experiments described in Figure 1 where TRDMT1-dependent sites of m5C modification after hydrogen peroxide treatment is described, the authors state, "we selected the best ~10% of the TRDMT1-dependent and -independent m5C sites for comparison based off of coverage and total methylation," but no specific metrics for how 'best' is defined are given. Since this leads to a narrow interpretation of the data, there is a risk that this incorporated bias into the study.
2. Similarly, in data presented in Figure 2, listing the proteins (even in the supplement) that preferentially bound unmodified, 5mC-modified, or both RNA:DNA hybrids in rank order would strengthen the argument that focusing on the alt-EJ pathway due to PARP1, XRCC1, and Lig3 is appropriate, if proteins from no other DNA repair pathway are significantly represented. If not, then confirmation bias cannot be ruled out.
3. The text describing Figure 3B says that m5C formation is lost in the TRDMT1 knockout, but it is only reduced 2-fold.
4. In the 2020 paper, the authors state that TRDMT1 and RAD52 are epistatic, but as both encoded enzymes have pleiotropic effects because they are involved in different pathways, this is a difficult point to argue. The reason this is relevant for this manuscript is that they use this as evidence to implicate the TRDMT1 knockout as driving alt-EJ. I'm surprised they never considered the possibility of the involvement of single-strand annealing. As SSA is a completely RAD52 dependent process, lack of RAD52 recruitment would severely affect SSA efficiency.
5. I'm confused about the design of the experiment represented in Figure 5F. The authors already have U2OS cells with a TRDMT1 knockout that they can use in the EJ2-GFP assay for alt-EJ in combination with Olaparib, why do they add variability by switching to an siRNA approach?
6. The 2021 paper describing the synthesis of the YW-1842 compound doesn't discuss if there is any cross-reactivity between TRDMT1 and similar enzymes. The only evidence shown are proliferation assays, which wouldn't indicate if the drug targets other methyltransferases that would affect the results.
7. A major suggested goal is that the TRDMT1 inhibitor could be given in combination with PARP inhibitors in cancer treatment. According to the data, where half of the top 10% are implicated in DSB-induced R-loops, would this only target a small subset of sites for inefficient DSB repair compared to a global defect in BRCA1/2-deficient cancer? Also, from a clinical perspective, PARP inhibitors are usually not given in combination due to a high level of adverse side effects and variable targeting of PARP1, PARP2, and PARP3, so I'm unsure of the strength of this argument. Results shown in Figure 6F and G show that PARPi alone without the TRDMT1 inhibitor induced significant cell death even in cells with active BRCA1/2.

Point-by-point responses to reviewers' comments

Reviewer #1 (Remarks to the Author):

Yang et al. propose a follow up work starting from the premises of their 2020 paper, demonstrating that TRDMT1-mediated m5C RNA modification at transcribed loci not only promotes HR repair of DNA damage, but it also prevents PARP-mediated Alt-NHEJ engagement.

First, they investigate the localization of TRDMT1-dependent m5C modification relatively to R-loops formation and sequence context. Subsequently, they identify proteins that bind differentially to unmodified and m5C modified hybrids, and they discover PARP among them. They use their published cell model to demonstrate that PARP enrichment to the site of damage is prevented by TRDMT1-dependent RNA methylation, and this is reflected by a boost in alt-NHEJ engagement and alt-NHEJ factors recruitment to the site of DNA damage in the absence of TRDMT1. Finally, they observe an interesting synthetic lethality using TRDMT1 inhibitors and alt-NHEJ factors inhibitors in cancer cells.

This is an overall interesting and solid paper, and I only have some concerns about the first set of results and the statistics used throughout the manuscript.

We sincerely appreciate the reviewer for their positive comments. We have carefully reviewed their feedback and made the necessary changes as outlined below.

The authors propose that TRDMT1-mediated RNA m5C formation primarily occurs in R-loops in the genome. To support this claim, they combine data of bisulfite-seq and DRIP-seq in H₂O₂-treated and unchallenged cells respectively. Next, they check the formation of damage-induced R-loops 2 out of 3 loci investigated as positive for m5C but negative for DRIP signal. I find these data a bit weak to reach their conclusions, as the presence of R-loops and the deposition of m5C was never investigated in the same exact condition. DRIP-seq in H₂O₂-treated cells could be beneficial to support authors' point, ideally showing a more specific co-presence of m5C and R-loops after DNA damage induction.

Response: We conducted DRIP-seq experiments using cells treated with H₂O₂. However, we did not observe a significant change in overlap between RNA m5C and DRIP peaks, which can be attributed to the fact that H₂O₂ induces DNA damage throughout the genome in a random distribution pattern that may not be consistent across all samples. To address this limitation, we decided to perform RNA m5C sequencing before and after inducing DNA damage with H₂O₂. We then compared the m5C sequencing results with the DRIP-seq data obtained after the damage. Interestingly, our findings revealed an increased overlap between m5C sites and DRIP-seq peaks following H₂O₂-induced damage (**Fig. 1A** on the right side). This result provides supportive evidence for the specific co-occurrence of m5C modifications and R-loops following DNA damage induction.

Fig. 1A The number of RNA m5C sites overlap with the DRIP-seq peaks.

Figure 1G: DRIP-qPCR experiment should include RNaseH negative controls to ensure that the differences in signal are representative of the modulation in hybrids formation

Response: We sincerely thank the reviewer for bringing this to our attention. In response, we have incorporated a control group with RNaseH treatment. Following the RNaseH treatment, we observed a decrease in DNA:RNA hybrids at the m5C-R, m5C1, m5C2, and m5C3 sites. We have replaced the **Fig. 1G** with this new result.

Fig. 1G. The levels of DNA:RNA hybrids in different genomic loci were analyzed by DRIP-qPCR with or without RNaseH treatment.

Figure S2C: it is unclear, in text authors report a comparison between the best TRDMT1-dependent m5C sites and the overall m5C, while, in the figure the bar is labeled as unspecified "Top 32 methylated C sites", and the caption reports "The fraction of top 32 methylated sites or bottom 32 unmethylated sites contained in the DRIP peaks".

Moreover, I am not certain that the comparison between the best TRDMT1-dependent m5C sites (defined in terms of coverage and methylation levels) and overall m5C is meaningful: the greater percentage of TRDMT1-dependent m5C sites colocalizing with DRIP peaks could be due to the fact that these "best sites" are more methylated (as they were selected based on that), rather than because they are TRDMT1-dependent.

Response: Thank you for pointing out the ambiguity in the text and figure. We apologize for any confusion caused by labeling. The label "Top 32 methylated C sites" are actually "Top 32 TRDMT1-dependent methylated C sites". The revised label helps to clarify that the comparison is specifically focused on the subset of m5C sites that are both methylated and dependent on TRDMT1. Consequently, the higher percentage of TRDMT1-dependent m5C sites colocalizing with DRIP peaks can be attributed to their elevated methylation levels, providing evidence of TRDMT1-dependent methylation. To ensure consistency and accuracy, all labels have been corrected to read "top TRDMT1-dependent methylated C sites."

Figure 2: in panel C and D authors could indicate the complexes of PARP+ substrate and the unbound substrate with arrows. Panel E lacks statistics

Response: Thanks for the comment. The panels are now appropriately labeled.

Additionally, statistical analysis by ANOVA has been included specifically for panel E to provide a comprehensive interpretation of the data.

Figure 4: all the comparisons in the figure should be tested by ANOVA, not t-test, which is used when the comparison is only between two samples.

Response: The statistical analysis for **Fig. 4** has been performed with ANOVA accordingly.

Figure 5: Authors could add a representative image also for Pol θ , as they did for XRCC1 and Lig3. Again, in figure 5F, the wrong statistical test was performed.

Response: Thanks for the reviewer's comment. The representative images for Pol θ have been included to provide visual representation (**Fig. 5C**). Furthermore, the statistical test for **Fig. 5F** has been rectified and revised accordingly.

Fig. 5C. WT and TRDMT1 KO U2OS-TRE cells transfected with TA-KR plasmid were light irradiated and allowed to recover for 0.5 h before fixation. Cells were stained with Pol θ antibody. Representative images are showed (scale bar: 10 μ m).

Figure 6: use of t-test instead of ANOVA throughout the figure

Response: The statistical analysis for **Fig. 6** has been performed with ANOVA accordingly.

Reviewer #2 (Remarks to the Author):

Bisulfite sequencing has been widely used to identify m5C modification in DNA and RNA. However, this technique suffers some drawbacks mainly due to the fact that sodium bisulfite-mediated deamination of C to U occurs successfully only on ssRNA. Therefore, potentially false positive coming from RNA secondary and higher order structures (e.g. GC-rich regions) might affect the correct identification of m5C sites. Furthermore, due to technical artifacts and lack of proper control, the m5C mapped sites can widely range from 100 to 12000 in mammalian genome.

Response: We appreciate the valuable suggestions provided by the reviewer. In response to the comments, we have conducted additional experiments to address the concerns raised. Specifically, we performed RNA m5C sequencing using bisulfite sequencing with and without hydrogen peroxide (H₂O₂) treatment in both wild-type (WT) cells and TRDMT1 knockout (KO) cells. In our newly generated **Fig. 1A**, we observed a basal level of RNA m5C in cells without any damage exposure. Upon induction of DNA damage, we observed an increase in the levels of methylated cytosine (m5C) in mRNA. This finding suggests that DNA damage leads to an elevation in RNA m5C levels. In TRDMT1 KO cells, both before and after damage induction, we only detected an equivalent basal level of RNA m5C. This suggests that the induction of TRDMT1-dependent RNA m5C is specifically triggered by DNA damage. The concern on the method will be further addressed below.

Fig. 1A. The number of RNA m5C sites identified from bisulfite sequencing in indicated cells before and after H₂O₂ treatment.

In this study, the identification of TRDMT1-dependent m5C sites has been determined in wt cells and TRDMT1 KO cells. However, the data from bisulfite sequencing should be repeated using catalytically inactive TRMMDT1 mutants, since compensatory roles of other m5C-methyltransferases cannot be excluded at this stage.

As reviewer suggested, we performed bisulfite sequencing in TRDMT1 KO cells that were transfected with a catalytically inactive mutant C79A TRDMT1 before and after DNA damage. Expression of this mutant in TRDMT1 KO U2OS-TRE cell lines was verified in previous publication² and confirmed in this study. We found that the catalytically inactive mutant of TRDMT1 in TRDMT1 KO cells showed decreased number of RNA 5mC sites after damage. We calculated the ratio of damage-induced RNA m5C by dividing the total number of RNA 5mC sites after damage by the number before damage. The Ratios are 3.12, 1.04, and 0.72 in WT cells, TRDMT1 KO cells, and TRDMT1 C79A cells, respectively. Data are deposited into NCBI SRA with the accession number PRJNA984695. The reviewer link is

<https://dataview.ncbi.nlm.nih.gov/object/PRJNA984695?reviewer=207jgr32ccphuc309mor005cvp>. The differential effects observed with the WT and mutant TRDMT1 strongly suggest the importance of TRDMT1's catalytic activity in mediating the damage-induced RNA m5C modification.

Furthermore, to exclude artifacts coming from GC-rich sequences, other m5C sequencing techniques such as TAWO-seq should be considered.

DRIP-seq employs the S9.6 antibody which binds RNA:DNA hybrids allowing the identification of the genomic fragments containing RNA:DNA hybrids. However, this technique has some crucial limitations including low resolution, no-strand specificity and the intrinsic bias of the S9.6 antibody toward GC-rich sequences (Bou-Nader Nat comm. 2022). Therefore, the analysis of the overlapping signal between the m5C and RNaseH-dependent RNA:DNA might be bias to the identification of m5C sites in GC-rich environment.

We attempted to perform the TAWO-seq technique in our laboratory but unfortunately encountered technical obstacles during our trials. Seeking guidance, we consulted with Dr. Chunxiao Song, the group known to have published the TAWO-seq technique¹. Unexpectly, we learned that this group is no longer performing this experiment, and we were unable to obtain guidance on the technique.

We agree that it is important to address concerns related to potential false positives resulting from RNA secondary and higher order structures, particularly in GC-rich regions, which could impact the accurate identification of m5C sites. To mitigate the impact of RNA secondary structures on m5C site identification, we have paid careful attention to sample preparation and experimental conditions. We tried to minimize any potential RNA secondary structure formation during the processing steps, such as RNA extraction, library preparation, and sequencing. Specically, we incorporated Heat denaturation (noted in methods), which is known to disrupt RNA secondary structures during sample preparations.

It should be noted that this detection flaw, if still present, would likely affect both TRDMT1 WT and KO cells to an equal extent before and after DNA damage. In our experimental results, we determined TRDMT1-dependent methylation sites under the same procedures and background (**Fig. 1**). Therefore, we believe that RNA secondary structures may not be the primary concern influencing our findings. We appreciate the importance of addressing the potential impact of RNA secondary structures and will continue to develop new methodologies and explore alternative approaches such as selective chemical labeling to ensure the robustness of our findings in the future.

RNA modifications such as m5C and m6A have can both promote HR via the modulation of DNA:RNA hybrid stability. It should be crucial to characterize the expression profile of m6A-writers in TRDMT1 ko cells, since the lack of TRDMT1 activity might change the expression level of m6A-writers and the deposition of m6A at damage sites and the choice of HR over NHEJ repair pathway. No data about the regulation of cell cycle and cell cycle progression in wt and TRDMT1 KO have been provided (there are no controls provided for TRDMT1 inhibitor).

Response: We appreciate the reviewer's suggestion. We have examined the protein levels of the m6A writers, METTL3 and METTL14. The Western blot analysis reveals that the protein levels of METTL3 and METTL14 remain unchanged in TRDMT1 knockout (KO) cells when compared to U2OS wild-type (WT) cells. This result is added to **New Supplementary Fig. 5B**.

In our previously published paper, we have provided evidence that the knockout (KO) of TRDMT1 does not have an impact on the cell cycle (Figure attached from cited publication) ². Furthermore, as depicted in **New Supplementary Fig. 7C** of this manuscript, treatment with a TRDMT1 inhibitor also does not induce any alterations in the cell cycle.

Furthermore, evaluation of down/up-regulated genes which can influence the choice of DNA damage repair pathway, cyclins expression, p53 and TGF-beta signaling is pivotal to claim the direct role of TRDMT1 and m5C role on the DNA repair pathway choice in vivo. Finally, since hydrogen peroxide can induce both ssDNA and dsDNA damage, would be informative show the gH2AX levels after H2O2 treatment.

Response: We appreciate the reviewer's suggestion. We examined the protein levels of Cyclin B1, Cyclin E, P53, and TGF-beta. The Western blot results indicate that the protein levels of these factors remain unchanged in TRDMT1 knockout (KO) cells compared to U2OS wild-type (WT) cells. This result is added to **New Supplementary Fig. 5C**.

Supplementary Fig. 5C. WT and TRDMT1 KO U2OS-TRE cells were collected for WB. Indicated proteins were blotted with corresponding antibodies.

We examined the γ H2AX level after H₂O₂ treatment. The WB result revealed an increase in γ H2AX levels after H₂O₂ treatment. This result is added to **New Supplementar Fig. 1A**.

Supplementary Fig. 1A. U2OS-TRE cells were treated with 1 mM H₂O₂ for 1 h before harvest for WB. The γ H2AX was blotted.

Figure 2 shows the binding affinity of PARP1 in presence of dsDNA and DNA:RNA hybrids. It would be beneficial to add the dissociation constant values of PARP1-dsDNA and PARP1-DNA:RNA hybrid.

Response: We have included the K_d (dissociation constant) values in the manuscript. The K_d values were calculated based on the curves in Fig 2C, 2D. The K_d value for the binding of PARP1 with double-stranded DNA (dsDNA) is determined to be 2.25 nM. Additionally, the K_d value for the binding of PARP1 with DNA:RNA hybrid is measured to be 4 nM. These values provide important insights into the affinity of PARP1 for these respective nucleic acid structures. The K_d values of PARP1-dsDNA and PARP1-DNA:RNA hybrid are recently reported in NAR and the values are 1.662 nM and 2.668 nM, respectively³, which are comparable to our result.

Furthermore, it is not clear whether other RNA-related damage modifications such as m⁶A and 8-OHG can influence PARP1 recruitment and PAR activity.

Thanks for reviewer's input regarding the influence of m⁶A on PARP1 recruitment and PAR activity at transcribed regions of the genome. Based on our previous study, we have found that m⁶A is not present at transcribed or nontranscribed regions of the genome, both before and after DNA damage² (Figure attached from cited publication). Consequently, we do not believe that m⁶A significantly impacts PARP1 recruitment and PAR activity in cells. Our findings indicate that the background levels of m⁶A are comparable when assessing PARP1 recruitment and PAR activity.

Supplementary Fig 2b from PMID: 32503981² is shown. U2OS-TRE cells were transfected with TA-KR or tetR-KR. The cells were stained with m6A antibody right after light activation. No m6A was detected.

We have also observed that 8-oxoG is induced equally at both transcribed (TA-KR) and nontranscribed regions (tetR-KR) of the genome with damage in our previous publication⁴ (Figure attached from cited publication). Furthermore, the presence of 8-oxoG is not detected before KR activation in both transcribed (TA-cherry) and nontranscribed regions (tetR-cherry). Despite the comparable background levels of 8-oxoG at tetR- and TA-KR, we have observed differential recruitment and activation of PARP1, indicating that 8-oxoG is not the determinant in this process. In contrast to this, the impact of RNA m5C on the recruitment and activation of PARP1 are observed both in cells but also in vitro.

Fig 2B from PMID: 24293652⁴ is shown: U2OS-TRE cells transfected with TA-KR or tetR-KR. The cells were stained with 8-oxoG antibody right after light activation. 8-oxoG is induced equally at TA-KR and tetR-KR of the genome with damage.

Figure 5 illustrates how the lack of TRDMT1 protein affects the choice of Alt-NHEJ over c-NHEJ. To confirm that TRDMT1-driven m5c deposition is responsible for the exclusion of PARP1 from DNA break in EJ2/EJ5-GFP reporter assays, TRDMT1 ko cells should be rescued with catalytically inactive C79A and R162A TRDMT1 mutants. Furthermore, DR-GFP reporter assay and SA-GFP reporter should be added to exclude the involvement of TRMMDT1 in other DNA repair pathways. Also no controls for reporter assays were provided (KD of known related proteins). To exclude indirect effects of the TRDMT1 absence on the switch off of Alt-NHEJ pathway further controls must be provided.

Response: Thanks for the reviewer for this suggestion. In response to the comment, we have incorporated the EJ2-GFP assay to evaluate the performance of TRDMT1 catalytically inactive mutants, specifically C79A and R162A. The results clearly demonstrate that unlike TRDMT1 wild-type (WT), which successfully restores the suppression of alternative non-homologous end joining (alt-NHEJ), both C79A and

R162A mutants fail to do so. This result is added to **New Fig. 5G**. Additionally, as a positive control for the EJ2-GFP assay, we presented in Figure 5F that the efficiency of alt-NHEJ repair is reduced following treatment with PARP inhibitors (PARPi). This control further validates the reliability and sensitivity of our experimental setup.

Fig. 5G. U2OS-TRE cells were transfected with EJ2-GFP and I-SceI-Cherry plasmids. The fraction of GFP-positive cells in the Cherry-positive population was analyzed by flow cytometry (n = 3, mean ± SD).

Fig. 5F. WT and TRDMT1 KO U2OS-TRE cells were transfected with EJ2-GFP and I-SceI-Cherry plasmids. The fraction of GFP-positive cells in the Cherry-positive population was analyzed by flow cytometry (n = 3, mean ± SD). Before and after plasmids transfection, the cells were also treated with or without 3 μM PARPi Olaparib

In both our previously published paper ² and this manuscript, we have demonstrated that the knockout (KO) of TRDMT1 does not have an impact on the non-homologous end joining (NHEJ) repair efficiency, as evidenced by the EJ5-GFP assay. Furthermore, as an additional control, we introduced the knocking down of 53BP1, which resulted in a decrease in NHEJ repair efficiency. This control further supports the validity of our findings and highlights the specific role of TRDMT1 in the observed outcomes. This result is added to **New Supplementary Fig. 5A**

Supplementary Fig. 5A. U2OS-TRE pretreated with 53BP1 siRNA or control siRNA were transfected with EJ5-GFP and I-SceI-Cherry plasmids. The fraction of GFP-positive cells in the Cherry-positive population was analyzed by flow cytometry (n = 3, mean ± SD).

In our previously published paper, we presented evidence that the knockout (KO) of TRDMT1 leads to a decrease in homologous recombination (HR) repair efficiency, as demonstrated by the DR-GFP assay ². In this manuscript, we performed the SA-GFP

assay and found that TRDMT1 KO does not have an impact on single-strand annealing (SSA) repair efficiency. To further support this observation, we introduced the knocking down of RAD52 as a control, which resulted in a decrease in SSA repair efficiency. This control reinforces the specificity of our findings and underscores the role of TRDMT1 in HR repair rather than SSA repair. This result is added to New Fig. 5H

Fig. 5H. WT and TRDMT1 KO U2OS-TRE cells pretreated with or without RAD52 siRNA were transfected with SA-GFP and I-SceI-Cherry plasmids. The fraction of GFP-positive cells in the Cherry-positive population was analyzed by flow cytometry (n = 3, mean ± SD).

Reviewer #3 (Remarks to the Author):

This work by Yang and colleagues is a follow-up to a 2020 study that examines the role of m5C modification of RNA and the subsequent effect on DNA repair pathway choice. The group has identified TRDMT1 as the enzyme responsible for m5C modification. Here, they demonstrate a correlation with regions that form transcriptionally coupled R-loops and sites of m5C modification of RNA following hydrogen peroxide treatment, meaning that these are sites where DNA damage-induced R-loops are forming and TRDMT1 is involved with m5C modification of RNA in an RNA:DNA hybrid. In the previous work, it was shown that this modification led to the recruitment of Rad52 to drive HR repair. Here, they show that when the ability to generate m5C modifications is blocked via a TRDMT1 deletion, PARP1 is recruited to RNA:DNA hybrids and activation and PARylation commences, meaning that the m5C modification driven by TRDMT1 is important in DSB repair pathway choice. It is suggested that pharmacological inhibition of TRDMT1 activity may drive cells to a PARP1-dependent repair pathway (alt-EJ) and subsequent inhibition of PARP will force the cells into a death pathway. While there are somewhat interesting results here implicating a novel factor with little-known prior function in a major repair pathway choice mechanism, it is difficult to evaluate the significance of the findings without better explanation of some of the results and additional experiments.

We sincerely appreciate the reviewer for their positive comments. We have carefully reviewed their feedback and made the necessary changes as outlined below.

1. For the experiments described in Figure 1 where TRDMT1-dependent sites of m5C modification after hydrogen peroxide treatment is described, the authors state, “we selected the best ~10% of the TRDMT1-dependent and -independent m5C sites for comparison based off of coverage and total methylation,” but no specific metrics for how ‘best’ is defined are given. Since this leads to a narrow interpretation of the data, there is a risk that this incorporated bias into the study.

Response: The “best” m5C sites were defined based on two criteria: high coverage (more than 25 reads) and high methylation levels (greater than 10%), which has been presented in the Methods section. These criteria were used to identify the most reliable and significant m5C sites.

Except for the sequence feature analysis in Fig. 1B and Fig. 1D which employed the best TRDMT-dependent m5C sites, There are also results which incorporated all identified m5C sites, like the distribution analysis of m5C in Fig. 1F. To avoid the misleading interpretation, we deleted the sentence “we selected the best ~10% of the TRDMT1-dependent and -independent m5C sites for comparison based off of coverage and total methylation.” and stated clearly in every panel what was used for analysis.

2. Similarly, in data presented in Figure 2, listing the proteins (even in the supplement) that preferentially bound unmodified, 5mC-modified, or both RNA:DNA hybrids in rank order would strengthen the argument that focusing on the alt-EJ pathway due to PARP1, XRCC1, and Lig3 is appropriate, if proteins from no other DNA repair pathway are significantly represented. If not, then confirmation bias cannot be ruled out.

Response: As requested by the reviewer, we have compiled a list of DNA repair proteins preferentially bound to unmodified, 5mC-modified, or both RNA:DNA hybrids in rank order (Supplementary Table 3). Among the proteins that exhibit a preference for unmodified hybrids, PARP1, XRCC1, and Lig3 are listed as the top three based on absolute intensity. Additionally, several ribosome proteins also display a preference for unmodified hybrids. The raw data and search files for the DNA:RNA hybrid pulldown fraction mass spectrometry have been deposited into the MassIVE data repository. The reviewer can access the data using the following link: <https://massive.ucsd.edu/ProteoSAFe/static/massive.jsp>. To log in as the reviewer, please use the username MSV000089337_reviewer and the password lanlab4406. Furthermore, when considering all other proteins bound to unmodified hybrids, PARP1 remains the top protein, while Lig3 and XRCC1 are ranked at No. 13 and No. 24, respectively, in the overall binding profile. This finding supports the notion that the Alt-NHEJ pathway is primarily affected.

Supplementary Table 1. DNA repair proteins identified in the DNA:RNA hybrid pulldown fraction mass spectrometry analysis

		Average Intensity/Relative Intensity		
		Empty control	Unmodified hybrid	m5C modified hybrid
Unmodified hybrids preferential binding	PARP1	2.70E+08/1	3.70E+09/13.7	1.50E+09/5.6
	LIG3	3.20E+05/1	3.40E+07/106.3	1.20E+07/37.5
	XRCC1	1.60E+05/1	1.50E+07/93.8	3.90E+06/24.4
	XRCC5	8.00E+05/1	6.50E+06/8.1	2.70E+06/3.4
	XRCC6	1.50E+06/1	6.30E+06/4.2	3.20E+06/2.1
	DDX1	3.00E+04/1	5.90E+06/196.7	2.30E+06/76.7
	RPA1	0	3.90E+06/2.2	1.80E+06/1
	SUB1	6.40E+04/1	1.40E+06/21.9	9.00E+05/14.1
m5C modified hybrids preferential binding	HIST1H2BB	2.20E+07/1	2.30E+07/1	4.50E+07/2
	NAMPT	0	0	6.90E+04/1
	PKN2	0	0	5.80E+04/1
	RMI1	0	0	2.20E+04/1
Equal binding	RFC3	1.20E+05/1	6.80E+05/5.7	6.90E+05/5.8
	RPA3	0	9.70E+05/1	1.40E+06/1.4
	RPA2	0	6.10E+05/1.3	4.60E+05/1

3. The text describing Figure 3B says that m5C formation is lost in the TRDMT1 knockout, but it is only reduced 2-fold.

Response: We appreciate the reviewer's observation and feedback. We have revised and clarified the text describing Fig. 3B to accurately reflect the findings. While it was

previously stated that m5C formation is lost in the TRDMT1 knockout, it has been corrected to indicate that it is actually reduced by approximately 2-fold in the absence of TRDMT1. Thank you for bringing this to our attention.

4. In the 2020 paper, the authors state that TRDMT1 and RAD52 are epistatic, but as both encoded enzymes have pleiotropic effects because they are involved in different pathways, this is a difficult point to argue. The reason this is relevant for this manuscript is that they use this as evidence to implicate the TRDMT1 knockout as driving alt-EJ. I'm surprised they never considered the possibility of the involvement of single-strand annealing. As SSA is a completely RAD52 dependent process, lack of RAD52 recruitment would severely affect SSA efficiency.

Response: In this manuscript, our findings from the SA-GFP assay indicate that the knockout (KO) of TRDMT1 does not have an impact on the single-strand annealing (SSA) repair efficiency. Additionally, to provide a control experiment, we introduced the knocking down of RAD52, which resulted in a decrease in SSA repair efficiency. These results further support our conclusion that TRDMT1 specifically influences homologous recombination (HR) repair rather than SSA repair. This result is added to **New Fig. 5H**.

Fig. 5H. WT and TRDMT1 KO U2OS-TRE cells pretreated with or without RAD52 siRNA were transfected with SA-GFP and I-SceI-Cherry plasmids. The fraction of GFP-positive cells in the Cherry-positive population was analyzed by flow cytometry (n = 3, mean ± SD).

5. I'm confused about the design of the experiment represented in Figure 5F. The authors already have U2OS cells with a TRDMT1 knockout that they can use in the EJ2-GFP assay for alt-EJ in combination with Olaparib, why do they add variability by switching to an siRNA approach?

Response: we agree with the reviewer's suggestion and we repeated the experiment with TRDMT1 KO cells. The result of TRDMT1 KO cell is similar with that of TRDMT1 knocking down and we replaced **Fig. 5F** with this new result.

Fig. 5F. WT and TRDMT1 KO U2OS-TRE cells were transfected with EJ2-GFP and I-SceI-Cherry plasmids. The fraction of GFP-positive cells in the Cherry-positive population was analyzed by flow cytometry (n = 3, mean ± SD). Before and after plasmids transfection, the cells were also treated with or without 3 μM PARPi Olaparib

6. The 2021 paper describing the synthesis of the YW-1842 compound doesn't discuss if there is any cross-reactivity between TRDMT1 and similar enzymes. The only evidence shown are proliferation assays, which wouldn't indicate if the drug targets other methyltransferases that would affect the results.

Response: Thank you for review reading the paper in 2021 and we appreciate your feedback. Regarding the specificity of YW1842, we want to clarify that we have included additional data addressing this aspect in our patent application, but not in the manuscript. We selectively show some data here rather than presenting it the manuscript because validating the specific of YW1842 is not the main goal for the manuscript. The compound is super selective against 190 kinase as well.

The effect of TRDMT1i was dependent on TRDMT1, but not tRNA methyltransferase or DNA methyltransferase. mRNA m5C dot blot was performed with methane blue as the loading control. mRNA was extracted from U2OS cells pretreated with different indicated siRNA. The cells were treated with or without TRDMT1i YW-1842.

7. A major suggested goal is that the TRDMT1 inhibitor could be given in combination with PARP inhibitors in cancer treatment. According to the data, where half of the top 10% are implicated in DSB-induced R-loops, would this only target a small subset of sites for inefficient DSB repair compared to a global defect in BRCA1/2-deficient cancer?

DSB-induced R-loops are transcription-dependent. In transcribed regions, where active gene expression occurs, the occurrence of DSBs can interfere with the transcription process and result in transcriptional abnormalities. If the DSBs are not promptly repaired, transcriptional machinery may stall leading to disruptions in gene expression, genome instability, and cell death. Therefore, although transcription-dependent repair occurs at the small percentage genome, it has been shown in previous studies that the lack of efficient DSB repair within transcribed regions of the

genome can have particularly significant and detrimental consequences on gene expression, genomic stability, and cell survival^{5, 6}.

Cancer cells are under increased transcriptional activity and dependency on sustained and dysregulated gene expression patterns, leading to a phenomenon known as transcription addiction. Consequently, any disruption or impairment in the transcription process can have profound consequences on the behavior and survival of cancer cells. In summary, the deficiency in DSB repair within transcribed regions of the genome can have severe effects, particularly in cancer cells due to their transcription addiction. Therefore, understanding and targeting the vulnerabilities associated with transcription addiction and DNA repair defects hold promise for developing novel therapeutic strategies against cancer⁷.

Also, from a clinical perspective, PARP inhibitors are usually not given in combination due to a high level of adverse side effects and variable targeting of PARP1, PARP2, and PARP3, so I'm unsure of the strength of this argument.

Thanks for the reviewer's comment. It is true that PARP inhibitors can have adverse side effects, as with any therapeutic agent, their safety profiles have been well-studied and generally well-tolerated in clinical trials. Recently, combination therapies involving PARP inhibitors have been extensively explored and have shown promising results in various clinical settings. Combination therapies involving PARP inhibitors can take different forms, such as combining PARP inhibitors with chemotherapy, radiation therapy, immunotherapy, or other targeted agents. These combinations are designed to exploit synergistic interactions between different treatment modalities, potentially leading to improved outcomes and overcoming resistance mechanisms⁸. For example, In the Phase III SOLO-1 trial, olaparib was combined with chemotherapy in newly diagnosed advanced ovarian cancer patients with BRCA mutations. The study showed that the combination significantly improved progression-free survival compared to chemotherapy alone. Moreover, in a Phase II clinical trial, niraparib was combined with pembrolizumab (an immune checkpoint inhibitor) in patients with metastatic triple-negative breast cancer. The combination demonstrated promising anti-tumor activity, with an objective response rate of 25%. Veliparib (a PARPi) + Temozolomide: In a Phase I/II clinical trial, the combination of veliparib with temozolomide (a chemotherapy agent) was investigated in patients with metastatic melanoma (clinicaltrials.com). The study demonstrated improved progression-free survival and overall survival compared to temozolomide alone. These examples represent a fraction of the numerous ongoing studies and clinical trials exploring the combination of PARP inhibitors with various treatment modalities in different cancer types.

Results shown in Figure 6F and G show that PARPi alone without the TRDMT1 inhibitor induced significant cell death even in cells with active BRCA1/2.

We used MDAMB231, a commonly studied breast cancer cell line that is often used as a model for triple-negative breast cancer (TNBC), which is known to be an aggressive and heterogeneous subtype of breast cancer. Although this cell line is

BRCA1/2 WT, studies have shown that MDAMB231 cells exhibit moderate sensitivity to PARPi alone⁹. While HRD is a key biomarker, it is not the sole determinant of response to PARPi. It is known that replication stress, activity of alternative DNA repair mechanisms also affect PARPi sensitivity¹⁰.

References

1. Yuan F, Bi Y, Siejka-Zielinska P, Zhou YL, Zhang XX, Song CX. Bisulfite-free and base-resolution analysis of 5-methylcytidine and 5-hydroxymethylcytidine in RNA with peroxotungstate. *Chemical communications* **55**, 2328-2331 (2019).
2. Chen H, *et al.* m(5)C modification of mRNA serves a DNA damage code to promote homologous recombination. *Nature communications* **11**, 2834 (2020).
3. Laspata N, *et al.* PARP1 associates with R-loops to promote their resolution and genome stability. *Nucleic acids research* **51**, 2215-2237 (2023).
4. Lan L, *et al.* Novel method for site-specific induction of oxidative DNA damage reveals differences in recruitment of repair proteins to heterochromatin and euchromatin. *Nucleic acids research* **42**, 2330-2345 (2014).
5. Bader AS, Hawley BR, Wilczynska A, Bushell M. The roles of RNA in DNA double-strand break repair. *British journal of cancer* **122**, 613-623 (2020).
6. van den Heuvel D, van der Weegen Y, Boer DEC, Ogi T, Luijsterburg MS. Transcription-Coupled DNA Repair: From Mechanism to Human Disorder. *Trends in cell biology* **31**, 359-371 (2021).
7. Bradner JE, Hnisz D, Young RA. Transcriptional Addiction in Cancer. *Cell* **168**, 629-643 (2017).
8. Ren N, *et al.* Efficacy and Safety of PARP Inhibitor Combination Therapy in Recurrent Ovarian Cancer: A Systematic Review and Meta-Analysis. *Frontiers in oncology* **11**, 638295 (2021).
9. Keung MY, Wu Y, Badar F, Vadgama JV. Response of Breast Cancer Cells to PARP Inhibitors Is Independent of BRCA Status. *Journal of clinical medicine* **9**, (2020).
10. Cybulla E, Vindigni A. Leveraging the replication stress response to optimize cancer therapy. *Nature reviews Cancer* **23**, 6-24 (2023).

REVIEWER COMMENTS

Reviewer #1 (Remarks to the Author):

The authors of the manuscript addressed all my questions, and their additional experiments improved the manuscript. Just a few observations:

In new figure 1G, right panel, statistics to show increased levels of DNA:RNA hybrids upon H₂O₂ treatment may be added, if appropriate.

I am still not convinced by the comparison in figure S2C: comparing the colocalizations with hybrids of TRDMT1-dependent highly methylated regions vs generic low methylated regions doesn't really tell us anything about TRDMT1, only that highly methylated regions display high levels of hybrids. A more informative comparison could be the colocalization between hybrids and top 10% TRDMT1-dependent m5C sites vs top 10% TRDMT1-independent m5C sites, if the authors want to make a point that TRDMT1-dependent methylation is preferentially present at hybrids sites compared to TRDMT1-independent methylation. Otherwise, if their point is to show that hybrids are preferentially present at sites of m5C formation, they could compare colocalizations of highly methylated vs low methylated sites with hybrids, regardless of TRDMT1 dependency. They can also simply show that half of the top 10% TRDMT1-dependent m5C colocalize with DRIP peaks, but the comparison that is shown does not add much information in my opinion.

Reviewer #2 (Remarks to the Author):

The authors responded to my previous comments in a satisfactory manner. However, there are a few outstanding issues that need to be addressed before considering this manuscript for publication in Nat Com.

The aim of this paper is to unveil the role of m5C modification on R-loops upon DNA damage. While this is an interesting point, there is a fundamental bias in the interpolation of the data from bisulfite-seq and DRIP-seq presented here. The authors first select the top 20 m5C sites within GC-rich regions and then correlate them to S9.6 peaks. This strategy leads to a biased conclusion that R-loops are m5C modified upon damage, given that S9.6 preferentially binds to GC-rich R-loops and the m5C deposition window has already been narrowed down to GC-rich sequences.

To exclude this possibility, the authors should overlap the total differentially expressed RNaseH-sensitive DRIP-seq peaks with the total m5C sites and check if their claims hold significance. At this stage, the authors cannot exclude the possibility that m5C deposition has a role in specific DNA damage contexts (e.g., inhibition of DNA topoisomerase I induced by H₂O₂) or only in minor DNA repair pathways such as alt-NHEJ. Therefore, the title "The RNA m5C modification in R-loops as an off switch of alt-NHEJ" is too strong and not fully supported by the data.

Moreover, the authors have partially addressed the concerns regarding the probable presence of false positives in the bisulfite-seq. However, the addition of the heat-snap cooling step might have had a positive impact on the identification of m5C modification within structured RNAs. I am wondering why the deposition of TRDMT1-independent m5C is not visible after the induction of H₂O₂ in TRDMT1 KO cells, given that the deposition of m5C has been described as both TRDMT1-dependent and independent. A dot blot on the m5C deposition would be helpful to assess the total m5C deposition upon damage, comparing conditions with and without damage in WT and TRDMT1 KO cells.

Reviewer #3 (Remarks to the Author):

The authors did their due diligence in addressing my comments and added several important experiments. In light of this, I am more supportive of publication.

Reviewer #1 (Remarks to the Author):

The authors of the manuscript addressed all my questions, and their additional experiments improved the manuscript. Just a few observations:

In new figure 1G, right panel, statistics to show increased levels of DNA:RNA hybrids upon H₂O₂ treatment may be added, if appropriate.

Response: We sincerely appreciate the reviewer for the positive comments. For Fig 1G right panel, the ANOVA statistics has been added as suggested, which showed increased levels of DNA:RNA hybrids upon H₂O₂ treatment

I am still not convinced by the comparison in figure S2C: comparing the colocalizations with hybrids of TRDMT1-dependent highly methylated regions vs generic low methylated regions doesn't really tell us anything about TRDMT1, only that highly methylated regions display high levels of hybrids. A more informative comparison could be the colocalization between hybrids and top 10% TRDMT1-dependent m5C sites vs top 10% TRDMT1-independent m5C sites, if the authors want to make a point that TRDMT1-dependent methylation is preferentially present at hybrids sites compared to TRDMT1-independent methylation. Otherwise, if their point is to show that hybrids are preferentially present at sites of m5C formation, they could compare colocalizations of highly methylated vs low methylated sites with hybrids, regardless of TRDMT1 dependency. They can also simply show that half of the top 10% TRDMT1-dependent m5C colocalize with DRIP peaks, but the comparison that is shown does not add much information in my opinion.

Response: We sincerely thank the reviewer for their valuable suggestion and apologize for any lack of clarity in our previous communication. In our previously published paper, we successfully demonstrated the occurrence of TRDMT1-dependent RNA m5C modification within the context of R-loops using the DART assay in cellular systems. To expand and validate this observation more comprehensively at the genome-wide level, we utilized a bisulfite sequencing approach as the distribution of RNA m5C was not thoroughly studied in previous studies. In Fig S2C, we present compelling evidence to convince the reader that TRDMT1-dependent RNA m5C sites exhibit a notable preference for the R-loop (DNA:RNA hybrid) context. Following the reviewer's recommendation, we additionally compared the rate of overlapping with R-loops for TRDMT1-independent m5C sites. Our findings reveal that TRDMT1-dependent RNA m5C sites predominantly overlap with DRIP pulled down R-loops, in comparison to TRDMT1-independent RNA m5C and unmethylated Cs. However, a minor percentage of overlap between DRIP and TRDMT1-independent RNA m5C suggests that other backup enzymes might contribute to RNA m5C in addition to TRDMT1. We are thankful for the constructive feedback and we believe these findings will significantly contribute to the understanding of RNA m5C modifications and their association with R-loops in cellular processes.

Supplementary Fig. 2C. The fraction of top 32 TRDMT1-dependent methylated sites or top 42 TRDMT1-independent methylated sites or bottom 32 unmethylated sites contained in the DRIP peaks.

Reviewer #2 (Remarks to the Author):

The authors responded to my previous comments in a satisfactory manner. However, there are a few outstanding issues that need to be addressed before considering this manuscript for publication in Nat Com.

We sincerely appreciate the reviewer for the positive comments. We have carefully reviewed their feedback and made the necessary changes as outlined below

The aim of this paper is to unveil the role of m5C modification on R-loops upon DNA damage. While this is an interesting point, there is a fundamental bias in the interpolation of the data from bisulfite-seq and DRIP-seq presented here. The authors first select the top 20 m5C sites within GC-rich regions and then correlate them to S9.6 peaks. This strategy leads to a biased conclusion that R-loops are m5C modified upon damage, given that S9.6 preferentially binds to GC-rich R-loops and the m5C deposition window has already been narrowed down to GC-rich sequences.

To exclude this possibility, the authors should overlap the total differentially expressed RNaseH-sensitive DRIP-seq peaks with the total m5C sites and check if their claims hold significance. At this stage, the authors cannot exclude the possibility that m5C deposition has a role in specific DNA damage contexts (e.g., inhibition of DNA topoisomerase I induced by H₂O₂) or only in minor DNA repair pathways such as alt-NHEJ. Therefore, the title "The RNA m5C modification in R-loops as an off switch of alt-NHEJ" is too strong and not fully supported by the data.

Response: We sincerely appreciate the reviewer's valuable comment. To ensure the selection of reliable and significant m5C sites, we applied two specific criteria during our analysis, as detailed in the Methods section. The m5C sites were chosen based on high coverage (more than 25 reads) and high methylation levels (greater than 10%). In response to the suggestion, we conducted a total m5C sites overlap analysis. The results demonstrate a higher percentage of overlap between the total m5C sites and the RNaseH-sensitive DRIP-seq peaks when compared to the overlap with sequenced unmethylated C sites (Supplementary Fig. 2D).

Moreover, the authors have partially addressed the concerns regarding the probable presence of false positives in the bisulfite-seq. However, the addition of the heat-snap cooling step might have had a positive impact on the identification of m5C modification within structured RNAs. I am wondering why the deposition of TRDMT1-independent m5C is not visible after the induction of H₂O₂ in TRDMT1 KO cells, given that the deposition of m5C has been described as both TRDMT1-dependent and independent. A dot blot on the m5C deposition would be helpful to assess the total m5C deposition upon damage, comparing conditions with and without damage in WT and TRDMT1 KO cells.

Response: We appreciate the reviewer's comment and have taken it into consideration for our study. To clarify, the TRDMT1-independent m5C sites are defined as m5C sites with a higher methylation level in TRDMT1 KO cells compared to WT cells. Despite the fact that the total number of RNA m5C sites did not increase significantly after H₂O₂ treatment in TRDMT1 KO cells, it is important to note that there is still basal level of methylation sites that specifically occur in the H₂O₂ damage group but not in the without damage group for the TRDMT1 KO cells. In a dot blot analysis, RNA m5C is indeed increased in TRDMT1 KO cells after H₂O₂ treatment. However, it is still significantly lower compared to the methylation levels observed in U2OS WT cells. Indeed, it is well-established that PARP1 is the primary enzyme responsible for poly-ADP-ribosylation, while other PARP enzymes also play a role in this reaction. Similarly, in the case of RNA modification, TRDMT1 is considered to be one of the predominant enzymes responsible for major RNA m5C induction after damage. However, as in many biological processes, redundancy and complexity often exist, and it is possible that other yet-unknown enzymes could contribute to the RNA m5C modification. These potential backup enzymes may play a supplementary role or even have distinct functions in specific cellular contexts. We appreciate the reviewer's thoughtful comment, and further investigation into the contribution of other potential enzymes will be crucial for a comprehensive understanding of the RNA m5C modification landscape.

The m5C dot blot for the mRNA extracted from U2OS WT and TRDMT1 KO cells with or without H₂O₂ damage. The cells were treated with 1mM H₂O₂ for 1 h before extraction for mRNA with Dynabeads™ mRNA DIRECT™ Purification Kit. Methylene blue was used as the loading control.

REVIEWERS' COMMENTS

Reviewer #1 (Remarks to the Author):

I have no further comments, I am happy with the present version of the manuscript.

Reviewer #2 (Remarks to the Author):

The authors made attempt to address my concerns, however, I request further clarification before considering this manuscript for publication in Nature Com.

My specific comments:

1. Supplementary Figure 2D. I would like to see the bar chart representing all DRIP-seq RNaseH sensitive peaks as 100%. How many of them are methylated (%) vs unmethylated (%)? For example 20 vs 80%?
2. Methylation dot blot. These data need quantification and should be added to the manuscript. The difference between WT H₂O₂ and KO H₂O₂ does not seem to be significant. Therefore their conclusion that TRDMT1 is the predominant m⁵C RNA methyltransferase upon DNA damage is exaggerated. These two comments need to be addressed.

Reviewer #2 (Remarks to the Author):

The authors made attempt to address my concerns, however, I request further clarification before considering this manuscript for publication in Nature Com.

My specific comments:

1. Supplementary Figure 2D. I would like to see the bar chart representing all DRIP-seq RNaseH sensitive peaks as 100%. How many of them are methylated (%) vs unmethylated (%)? For example 20 vs 80%?

Response: We thank the reviewer's question.

The total count of DRIP-seq peaks sensitive to RNaseH exceeds 50,000, whereas the number of mRNA m5C sites identified from bisulfite sequencing is approximately one thousand. This substantial difference arises from limitations in current approaches for detecting modified RNA. Given the considerable disparity in baseline counts between methylation sites and DRIP-seq peaks, using the number of DRIP-seq peaks as a baseline will result in a low readout for comparison. We therefore conducted a comparative analysis between the top 32 methylated sites vs. bottom unmethylated sites overlapped with RNaseH -sensitive DRIP peaks. The results revealed that 40% methylated sites exhibited overlap with RNaseH-sensitive DRIP-seq peaks before damage whereas only 6% in bottom unmethylated sites displayed an overlapping with RNaseH -sensitive DRIP peaks. This difference underscores the trend of TRDMT1-dependent mRNA m5C sites preferentially occurs in the context of R-loops.

2. Methylation dot blot. These data need quantification and should be added to the manuscript. The difference between WT H₂O₂ and KO H₂O₂ does not seem to be significant. Therefore their conclusion that TRDMT1 is the predominant m5C RNA methyltransferase upon DNA damage is exaggerated.

Response: We thank the reviewer's suggestion and apologize that we didn't make it clear. We did the quantification as suggested. In the dot blot, RNA m5C is indeed increased in TRDMT1 KO cells after H₂O₂ treatment. However, it is still significantly lower compared to the methylation levels observed in U2OS WT cells. The data has been added to new Supplementary Fig. 2a.

Supplementary Fig. 2a. The m5C dot blot for the mRNA extracted from U2OS WT and TRDMT1 KO cells with or without H₂O₂ damage. The cells were treated with 1mM H₂O₂ for 1 h before extraction for mRNA with Dynabeads™ mRNA DIRECT™ Purification Kit. Methylene blue was used as the loading control. The levels of m5C was quantified (n = 3 independent experiments, Mean ± SEM). The Statistical analysis was done with one-way ANOVA